# A human lung alveolus-on-a-chip model of acute radiation-induced lung injury

Queeny Dasgupta[1,2], Amanda Jiang[1,2], Amy M. Wen[2], Robert J. Mannix[1], Yuncheng Man [1,2], Sean Hall[2], Emilia Javorsky[2] & Donald E. Ingber [1,2,3] ✉

Acute exposure to high-dose gamma radiation due to radiological disasters or cancer radiotherapy can result in radiation-induced lung injury (RILI), characterized by acute pneumonitis and subsequent lung fibrosis. A microfluidic organ-on-a-chip lined by human lung alveolar epithelium interfaced with pulmonary endothelium (Lung Alveolus Chip) is used to model acute RILI in vitro. Both lung epithelium and endothelium exhibit DNA damage, cellular hypertrophy, upregulation of inflammatory cytokines, and loss of barrier function within 6 h of radiation exposure, although greater damage is observed in the endothelium. The radiation dose sensitivity observed on-chip is more like the human lung than animal preclinical models. The Alveolus Chip is also used to evaluate the potential ability of two drugs - lovastatin and prednisolone - to suppress the effects of acute RILI. These data demonstrate that the Lung Alveolus Chip provides a human relevant alternative for studying the molecular basis of acute RILI and may be useful for evaluation of new radiation countermeasure therapeutics.

Acute radiation induced lung injury (RILI) can occur following exposure to high doses of ionizing radiation following accidental or deliberate nuclear incident[1]. RILI can also be a toxic side effect of radiation therapy, which is administered to ~60% of cancer patients[2]. Radiation injury in lung cells causes DNA damage and accumulation of reactive oxygen species (ROS), which induces an inflammatory response that causes alveolar epithelial cell damage and severe injury to the neighboring endothelium[3,4]. This inflammatory phase is called radiation-induced pneumonitis (RP), occurs between 1–3 weeks of exposure[3,5] and is commonly diagnosed based on clinical symptoms, including shortness of breath, dry cough, low-grade fever, chest pain or general malaise. However, no specific diagnostic tests or imaging approaches can definitively correlate findings to clinical symptoms[5]. At present, most patients are given prolonged therapy with glucocorticoids, such as prednisolone, to reduce inflammation over 4–12 weeks after exposure[6]. Unfortunately, there is limited evidence to support its effectiveness at reducing pneumonitis, although there can be some small benefits with regards to subsequent fibrosis[7]. There is, therefore, a great need for

clinically relevant preclinical models that may be used to better understand the molecular basis of acute RILI as well as for the development of improved diagnostics and radiation countermeasure therapeutics.

The plausible threat of radiological or nuclear attacks, has led to the establishment of Centers for Medical Countermeasures against Radiation (CMCRs) across the U.S.; one of the missions of this network is to identify and develop therapeutics that can be used to treat victims of a radiological event[8]. The development of medical countermeasures "to diagnose, prevent, protect from or treat conditions that result from radiation exposure"[9] has been identified as a key priority of the US Food and Drug Administration (US FDA). Given that clinical studies are not ethical or feasible in the context of radiation medical countermeasures, high-fidelity preclinical models are required to evaluate the efficacy of novel medical countermeasures[10].

Consequently, animal models have been employed to understand RILI, but their failure to mimic clinically relevant dose sensitivities and recapitulate key hallmarks of the human pathophysiology limit their value[11]. For example, mouse RILI models only exhibit low grade

[1]Vascular Biology Program and Department of Surgery, Boston Children's Hospital and Harvard Medical School, Boston, MA 02115, USA. [2]Wyss Institute for Biologically Inspired Engineering at Harvard University, Boston, MA 02215, USA. [3]Harvard John A. Paulson School of Engineering and Applied Sciences, Harvard University, Cambridge, MA 02139, USA. ✉e-mail: don.ingber@wyss.harvard.edu

pneumonitis and minimal pulmonary fibrosis[12–14], and typically use animal survival as the primary experimental endpoint[15], instead of incidence and severity of pneumonitis, which is more human-relevant. Currently, non-human primate (NHP) models are considered the gold-standard for radiation injury[16,17], but their use is limited by short supply, long breeding periods, high costs, and serious ethical concerns[18,19]. In vitro models comprising human lung epithelial cells exposed to radiation provide some insight to the effects of radiation on individual cell types, but do not provide more physiologically relevant information about inflammatory responses, destruction of tissue barriers, and the interplay between different cell types during RILI that are observed in the human body and central to the development of radiation-induced pneumonitis[20–22].

In the present study, we leveraged organ-on-a-chip (Organ Chip) microfluidic culture technology to model acute RILI in the human lung in vitro. Organ Chip models of lung alveolus and airway have been successfully used in the past to model multiple lung diseases, including asthma, chronic obstructive pulmonary disease (COPD), pulmonary thrombosis[23], cystic fibrosis[24], and lung cancer[25], as well as viral infection and evolution of resistance to antiviral therapy[26,27]. Here, we used a previously described 2-channel Lung Alveolus Chip containing primary human lung alveolar epithelial cells cultured under an air-liquid interface (ALI) interfaced across an extracellular matrix (ECM)-coated porous membrane with a primary human lung microvascular endothelium that lines a channel which is dynamically perfused with culture medium with or without circulating human immune cells[22,24,25]. The engineered alveolar-capillary interface experiences cyclic mechanical strain to mimic breathing motions. Importantly, we show that when this human Lung Alveolus Chip is exposed to clinically relevant doses of gamma radiation, many of the physiological hallmarks of acute RILI are observed. This model enables us to recapitulate RILI in response to clinically relevant, one-time, doses of radiation (up to 16 Gy), as might occur in a radiation disaster. This dose is also within the range of the high radiation dose exposures used in some targeted ablative radiation therapies for cancer[28]. This in vitro Lung Alveolus model also recapitulates the known effects of radiation on induction of cytoprotective gene, hemoxygenase-1 (HMOX1) and can be used to study the responses of known radiation countermeasure drugs (e.g., prednisolone) as well as potential modulators of HMOX1, to radiation exposure in the human lung.

## Results

### Modeling RILI in the human Lung Alveolus Chip

The Lung Alveolus Chip is a microfluidic culture device that contains two microchannels separated by a porous ECM-coated membrane lined by human primary alveolar epithelial cells on one side and human lung microvascular endothelial cells (HMVEC-L) on the other to mimic the alveolar-capillary interface of human lung (Fig. 1a). Immunostaining for ZO-1 in tight junctions and E-cadherin in cell-cell adhesions in alveolar epithelial cells and VE-cadherin in endothelial cell-cell adhesions, along with DAPI staining of nuclei, confirmed the presence of a continuous alveolar-capillary interface with the endothelium lining all four sides of the lower channel beneath the porous membrane with a monolayer of alveolar epithelium above (Fig. 1a and Supplementary Fig. S1a). The experimental protocol (Fig. 1b) involves culturing the cells for 14 days at an ALI in the presence of cyclic mechanical strain to mimic breathing motions in order to ensure optimal differentiation (see Methods) before exposure to gamma radiation on Day 15.

Ionizing radiation induces direct damage to the cells by generating DNA double-strand breaks (DSBs), leading to hyperphosphorylation of DSB-associated proteins, which produce discrete nuclear foci containing p53-binding protein 1 (53bp1)[29,30]. When we exposed the Alveolus Chips to increasing doses of radiation from 0 to 16 Gray (Gy), we observed a similar dose-dependent rise in DNA damage (DSB formation) in both the epithelium and endothelium, as measured by quantifying the number of 53bp1-positive foci per nucleus (Fig. 1c, d). Interestingly, DSB formation could be detected with levels of radiation exposures as low as 1 Gy (Fig. 1c, d).

While both the alveolar epithelium and endothelium formed monolayers composed of cells surrounded by continuous cell-cell junctions in control chips (Fig. 1a, e and Supplementary Fig. S1a), damage to the tight junctions and a reduction in cell viability (Supplementary Fig. S1b) were observed when the chips were exposed to the highest radiation dose (16 Gy), but not at the lower exposures that induced DNA damage (Fig. 1e versus Fig. 1d). This was further corroborated by quantifying tissue barrier function, which showed that the apparent permeability ($P_{app}$) of the alveolar-capillary interface increased by ~7 fold after 6 h exposure to 16 Gy radiation (Fig. 1f), and this was accompanied by visible accumulation of fluid in the epithelial channel between 5 to 7 days following radiation exposure in ~85% of the radiated chips. Thus, the Alveolus Chip recapitulates the compromise of lung tissue barrier function that is a critical feature of RILI, which also emerges about a week after exposure to a similar high dose of radiation in humans.

As inflammation is also a common feature of RILI, we analyzed protein expression levels of multiple proinflammatory cytokines in chips exposed to radiation. However, we could not detect any changes in key proinflammatory cytokines (IL-6, IL-8, TNF-α, GM-CSF) or leukocyte adhesion markers (ICAM-1, E-selectin) even at the highest 16 Gy dose (Supplementary Fig. S1c). This observation raised the possibility that the lack of this response was due to the absence of immune cells. To explore this possibility, we performed similar studies while perfusing the endothelium-lined vascular channel with primary human peripheral blood mononuclear cells (PBMCs) before and during exposure to radiation. Importantly, the presence of PBMCs at the time of radiation exposure caused a sustained and progressive inflammation response in the Alveolus Chip (Fig. 1g). We observed upregulation of IL-8, IL-1α, P-selectin, IL-6, PAI-1, MIP-1a, TNF-α, ICAM-1 at 1 day (Fig. 1g) and several other cytokines at 1 to 7 days after radiation exposure (Fig. 1h). While the inflammatory response was similar at 1 and 2 days, the Alveolus Chips displayed a heightened response to radiation in both expression levels and in the number of affected cytokines at day 7 (Fig. 1h). For example, IL-1β, IL-7, GM-CSF and TGF-β were only overexpressed at this later time point. These findings are consistent with the clinical observation that the onset of RP occurs about 1 week after injury[3]. We also observed that barrier permeability ($P_{app}$) was significantly higher in radiated samples when PBMCs were present compared to control (Fig. 1i) and this correlated with increased disruption of PECAM1-containing cell-cell junctions in the endothelium (Supplementary Fig. S1d). Owing to the importance of PBMCs to generate an inflammation response, all subsequent radiation experiments on the Alveolus Chip were performed in the presence of PBMCs.

Analysis of acute cellular responses to radiation in the presence of PBMCs revealed significant increases in ROS levels within 2 h (Fig. 2a) and this response was dose-dependent (Supplementary Fig. S1e, f). This was accompanied by hypertrophy of the alveolar epithelial cells at 6 h (Fig. 2b, c), which again mimics a feature commonly seen in alveolar injury in vivo[3,31]. While cellular hypertrophy was not detected in the endothelium at this early time point, the size of both the epithelial and endothelial cells increased significantly by day 7 (Fig. 2b, c). Furthermore, when we analyzed effects on alveolar Type I and II cells by quantifying expression of biomarkers for each - aquaporin 5 (AQP5) and surfactant protein B (SPB), respectively - we observed a significant decrease in the expression of the alveolar Type I marker with a concomitant increase in the Type II marker at 6 h after exposure to 16 Gy radiation (Supplementary Fig. S1g), although there was no significant change in expression at lower doses. This finding among others led us to focus the bulk of our studies using 16 Gy radiation.

We then investigated how radiation exposure influences PMBC recruitment over time. When immune cells were perfused through the

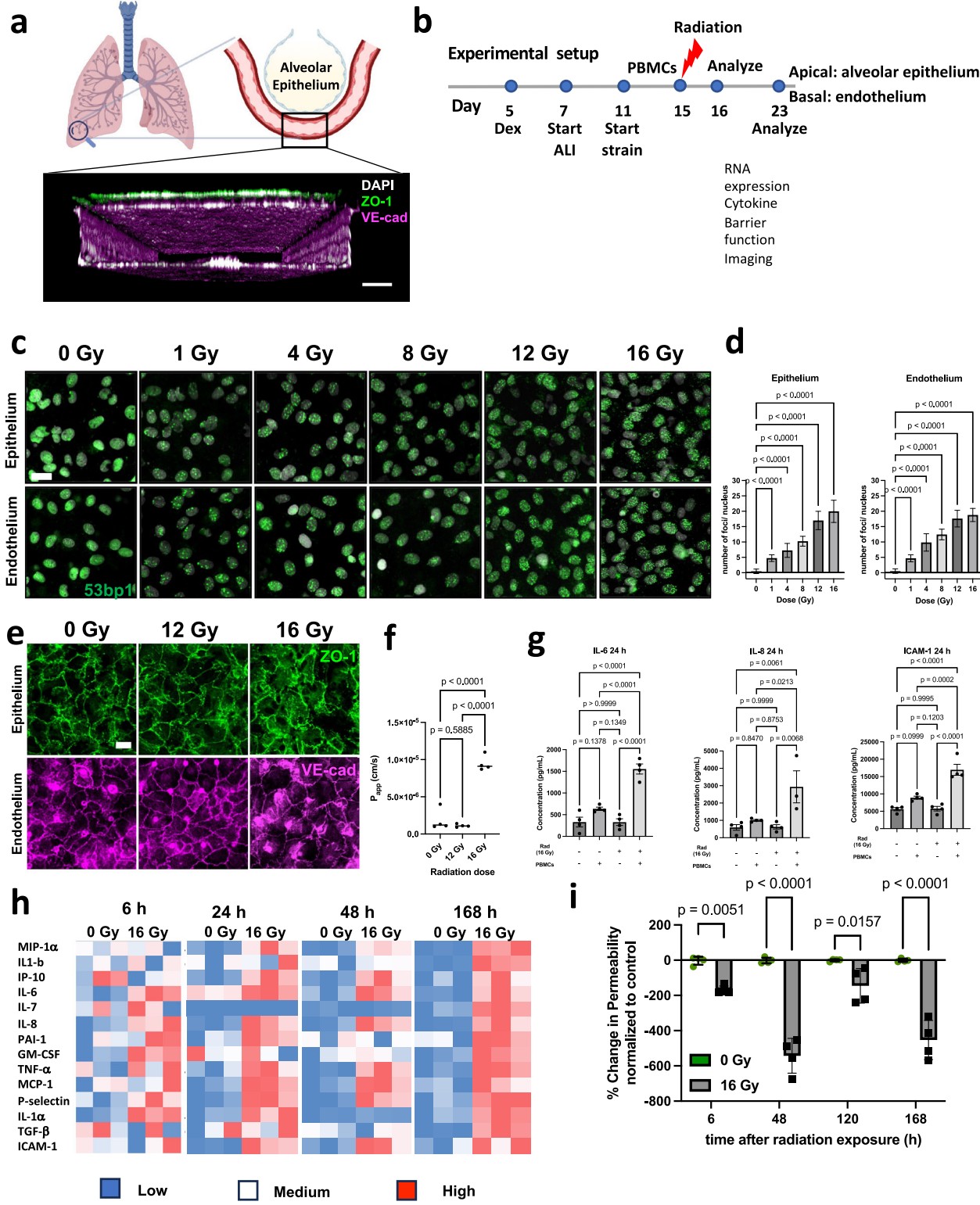

vascular channels of the Lung Alveolus Chip and exposed to 16 Gy radiation, 10 times more PBMCs bound to the endothelium compared to control non-radiated chips when analyzed 6 h later, and this recruitment increased by almost 5-fold more (levels in radiated chips ~46 times higher than control) by 7 days after radiation (Fig. 2d and Supplementary Fig. S1h).

Importantly, when similar studies were carried out by adding PBMCs to irradiated static Transwell co-cultures lined by human lung alveolar epithelium and microvascular endothelium interfaced across

a rigid, porous, ECM-coated membrane and cultured in the same medium, we did not observe any of these classic hallmarks of RILI. Transwell cultures failed to mimic the hypertrophy of the alveolar cells (Supplementary Fig. S2a, b), increase in barrier permeability (Supplementary Fig. S2c), or changes in gene expression (Fig. 2e) observed in response to radiation exposure in the Alveolus Chips. At the RNA level (Fig. 2e), the endothelium in these static cultures overexpress the leukocyte adhesion marker *ICAM-1* but did not display other gene expression changes that are characteristic of RILI in vivo which were

**Fig. 1 | Human Lung Alveolus Chip recapitulates hallmark features of RILI.**
**a** Schematic of the alveolus-on chip model (created with BioRender.com), showing the confocal z-stack illustrating endothelial tube formation. Scale bar = 100 µm. **b** Line diagram showing experimental plan, Dex, dexamethasone; ALI, air-liquid interface. **c** 53bp1 immunostaining (green) for double-stranded DNA breaks 2 h after radiation. DAPI counterstaining is shown in white. Scale bar = 20 µm. **d** Formation of 53BP1 foci was quantified per nucleus and showed a dose-dependent increase for both alveolar epithelial and endothelial cells. Data shown are mean +/− S.D. (analyzed nuclei per group $n = 50$ nuclei; one-way ANOVA DF = 5, $F = 525$, $p < 0.05$. **e** Immunostaining with ZO-1 (epithelial cells; green) and VE-cadherin junctions (endothelial cells; magenta) post irradiation, showed that junction disruption required a minimum dosage of 16 Gy. Scale bar = 20 µm.

**f** Barrier function assay showed a 7-fold increase in the apparent permeability co-efficient (Papp) at 6 h post radiation exposure to 16 Gy, but no difference in response to 12 Gy. One-way ANOVA: $n = 4$ chips in each condition, DF = 2, $F = 80.6$, $p < 0.05$. **g** Representative comparison of cytokine levels, 24 h post-radiation shows an inflammatory response to 16 Gy radiation, in the presence of PBMCs. One-way ANOVA, $n = 4$ chips for each condition, Data shown are mean +/− S.D. F(IL-6) = 40.05, F(IL-8) = 7.74, F (ICAM-1) = 34.07, $p < 0.05$. **h** Heatmap showing cytokine response to radiation at 6 h, 24 h, 48 h and 7 d post radiation exposure, in the presence of PBMCs, $n = 3$ chips for each condition, $p < 0.05$. **i** % change in barrier integrity normalized to 0 Gy control over 7 days post-radiation exposure, in the presence of PBMCs. 2-way ANOVA, $n = 4$ chips in each condition, DF = 3, $F = 19.3$, Data shown are mean +/− S.D. Significance at $p < 0.05$.

observed on-chip (e.g., increased expression of *HMOX1, ACTA2, DDIT3*, etc.). Similarly, while exposure of Transwell cultures to radiation induced an initial increase in cytokines like ICAM-1, MCP-1, IL-6 and IL-8 at 6 h post-radiation, the levels of these inflammatory cytokines returned to baseline levels by 24 h and there were no significant differences compared to controls (Fig. 2f). Furthermore, cells irradiated on Transwells exhibited a lower number of nuclear 53bp1 foci indicating reduced DNA damage in response to the same 16 Gy radiation exposure (Fig. 2g). Thus, these results demonstrate the importance of the ability of the microfluidic Lung Alveolus Chip to recreate the local physical microenvironment experienced by breathing lung alveolar tissues in vivo, including fluid flow and cyclic mechanical deformations (as well as an ALI), when attempting to model RILI in vitro.

Taken together, these studies established that the human breathing Lung Alveolus Chip can recapitulate key hallmarks of RILI in vitro, including DNA damage, junction disruption, increased barrier permeability and a heightened inflammatory response, whereas the same cells cultured under static conditions do not. Importantly, a radiation dose of 16 Gy and the presence of PBMCs were both necessary to capture this acute lung injury response on-chip.

## DNA damage and cell cycle arrest dominate early phase of RILI

We also carried out transcriptomic analyses to further understand the mechanistic pathways underlying acute RILI in the human Lung Alveolus Chip. At 6 h post radiation exposure, 11976 genes were detected in the epithelium and 19661 genes in the endothelium with 217 genes differentially expressed (37 upregulated and 180 downregulated) in chips exposed to 16 Gy versus controls (Fig. 3a, b). Interestingly, most genes were suppressed by exposure to radiation (Fig. 3a, b) and 85 of the 217 genes altered their expression in response to radiation exposure in both the epithelium and endothelium (Supplementary Fig. S3a); all but 3 of these common genes also were downregulated rather than being induced by radiation (Supplementary Fig. S3b, c). Unsupervised principal-component analysis (PCA) of RNA seq clustering data demonstrated two distinct gene expression clusters in both the epithelium and endothelium for radiated and not radiated cultures (Supplementary Fig. S3d, e). Gene set enrichment analysis (GSEA) of the hallmark pathways revealed downregulation of pathways involving the G2M checkpoint, E2F targets and mitotic spindle in the epithelium as well as endothelium (Fig. 3c, d, Supplementary Tables S1, S2), which is consistent with similar DNA damage to both lung tissues in response to radiation, as indicated by the 53bp1 staining (Fig. 1c, d). However, the endothelium also exhibited upregulation of immune response and allograft rejection pathways that generally represent a protective response to injury[22] (Fig. 3d). Gene ontology (GO) analysis similarly showed downregulation of cell division pathways relating to mitotic spindle formation, nuclear division, cytokinesis, and chromatid segregation in both lung tissue types (Supplementary Fig. S3f and Tabled S3 & S4). Our transcriptomic data also revealed that *EDN1, RUNX3, TGFB1*, and *CXCL8* genes are all upregulated in response to 16 Gy radiation (Fig. 3a, b), while there was a decrease in *PECAM1* expression and loss of endothelial cell-cell

junctions (Supplementary Fig. S1d). This is consistent with the induction of an endothelial-mesenchymal transition that has been previously described in models of radiation-induced injury[32], which causes endothelial cells to lose their polarity and become more migratory and invasive into surrounding tissues[33].

## Sustained inflammation is observed 7 d post-radiation

In contrast, when we analyzed similar responses at 7 days after radiation exposure, greater changes in gene expression were observed in the vascular endothelium compared to the alveolar epithelium (Fig. 4). At this later time point, 595 genes were differentially expressed in the epithelium in response to radiation with 1519 genes changed in the endothelium and 111 genes changing in both (Supplementary Fig. S4a). In contrast to the 6 h time point, more genes were upregulated than downregulated in both tissue types (Supplementary Fig. 4b, c). Unsupervised PCA of RNA seq clustering data again revealed 2 distinct gene expression clusters in both tissue types in radiated Alveolus Chips compared to controls (Supplementary Fig. S4d, e). Interestingly, GSEA demonstrated that the hallmark cell cycle pathways that were previously downregulated at 6 h were upregulated at 7 days (Fig. 4c, d, Supplementary Tables S5 and S6). This is consistent with the analysis of reactome pathways that showed an increase in DNA repair, including base excision repair, and activation of ataxia telangiectasia and Rad3-related protein (ATR) in response to replication stress, as well as an increase in senescence and senescence-associated secretory phenotype. GSEA analysis of the KEGG pathways also revealed upregulation of cytosolic DNA sensing in response to radiation in the epithelium (Fig. 4c), most likely mediated by induction of *ZBP1*[34]. Interferons are known to induce overexpression of *ZBP1*[35], and they are upregulated in response to radiation in the epithelium as well (Fig. 4a). Cytokine-cytokine receptor interactions and chemokine signaling pathways were also upregulated at this late time after exposure to radiation and this was accompanied by increases in expression of several inflammatory markers, including *IL6, IL8*, interferon Lambda 1 (*IFNL1*), and C-C Motif Chemokine Receptor 1 (*CCR1*), as well as general upregulation of interferon pathways, in the alveolar epithelium (Fig. 4a, c and Supplementary Fig. S4f, Supplementary Table S7).

In contrast, GSEA analysis of the vascular endothelium revealed significant upregulation of immune response, inflammatory response, and allograft rejection pathways mediated by TNF-α signaling via NF-κB, KRAS signaling, complement activation, IL2 STAT5 signaling, IL6 JAK-STAT3 and IFN-γ response pathways (Fig. 4b, d and Supplementary Fig. S4f, Supplementary Table S8). Analysis of reactome pathways confirmed that the NF-κB pathway was upregulated in the endothelium within the irradiated Alveolus Chip, and that this was accompanied by progressive and sustained inflammation marked by the activation of late-stage inflammatory markers *ILIB, TGFB1, IL10*, and *IL16*, together with an upregulation of *CXCL8, CXCR3*, and *CXCL10* (Fig. 4d and Supplementary Fig. S4e). We also observed increased expression of the potent vasoactive peptide endothelin-1 (*EDN1*) as well as oxidized low-density lipoprotein receptor-1 (*OLR1*), which is a key molecule associated with endothelial dysfunction during atherosclerosis progression.

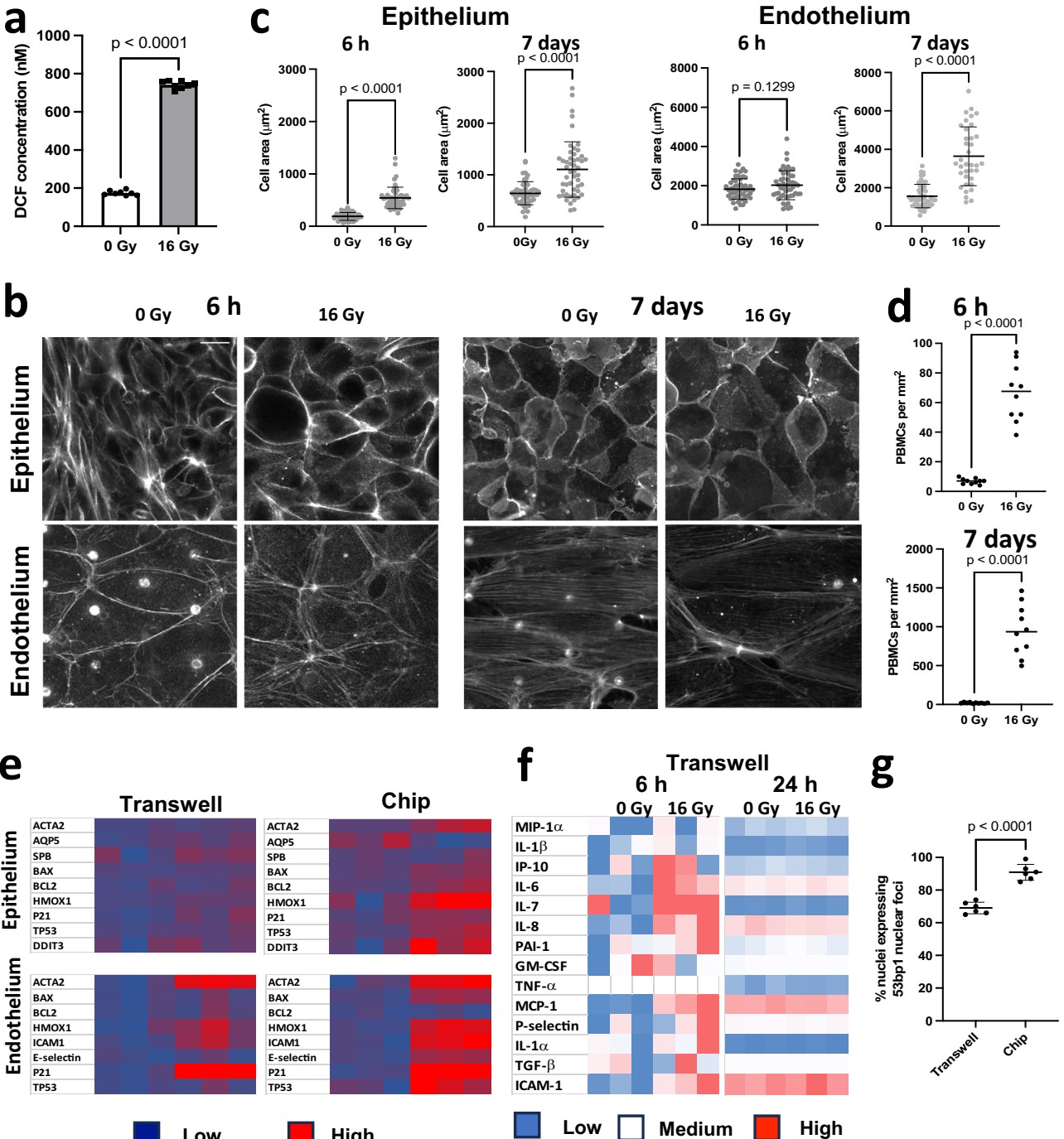

**Fig. 2 | Radiation injury causes ROS, cellular hypertrophy and increased recruitment of PBMCs over time on the alveolus chip. a** 16 Gy radiation induces increase in the ROS levels, 2 h after radiation injury, $n = 8$ chips, $F = 2.48$, DF = 7, Data shown are mean +/− S.D. $p < 0.001$. **b** IF images showing the effect of radiation on cellular hypertrophy in the epithelium and endothelium 6 h and 7 d post radiation exposure, Scale bar = 20 μm. Corresponding quantification of cellular area in the epithelium and endothelium in **c**, $t$-test with Welch's correction, where $n = 50$ cells in each condition, DF = 49, F (epi_6h) = 7.7, $p < 0.05$, F(epi_7d) = 5.7, F(endo_6h) = 9.8, F(endo_7d) = 8.3, Data shown are mean +/− S.D. **d** Quantification of PBMC recruitment to the vascular endothelium at 6 h and 7 d post-radiation exposure, $n = 10$ images taken from 3 biological replicate chips in each condition,

2-tailed unpaired $t$-test with F(6 h) = 104.4, F (7 d) = 2326, DF = 9, $p < 0.0001$. **e** Heatmaps comparing gene expression profiles in Transwells and chip ($n = 3$ transwells or chips), $p < 0.05$. The chip shows susceptibility to radiation that is evidenced by the overexpression of several markers. **f** Heatmap showing cytokine profiles of transwells at 6 h and 24 h post-radiation. At 6 h, the transwells exhibit an upregulation of IL-6, IL-8 and ICAM-1 but at 24 h, no effect of radiation exposure is observed. $n = 3$ transwells. **g** Quantification of % nuclei expressing nuclear foci in response to radiation in transwells vs chips. Data shown are mean +/− S.D. Two-tailed Student's $t$ test with Welch's correction, $n = 6$ frames from 2 biological replicates, $F = 1.88$, DF = 5, $p < 0.0001$.

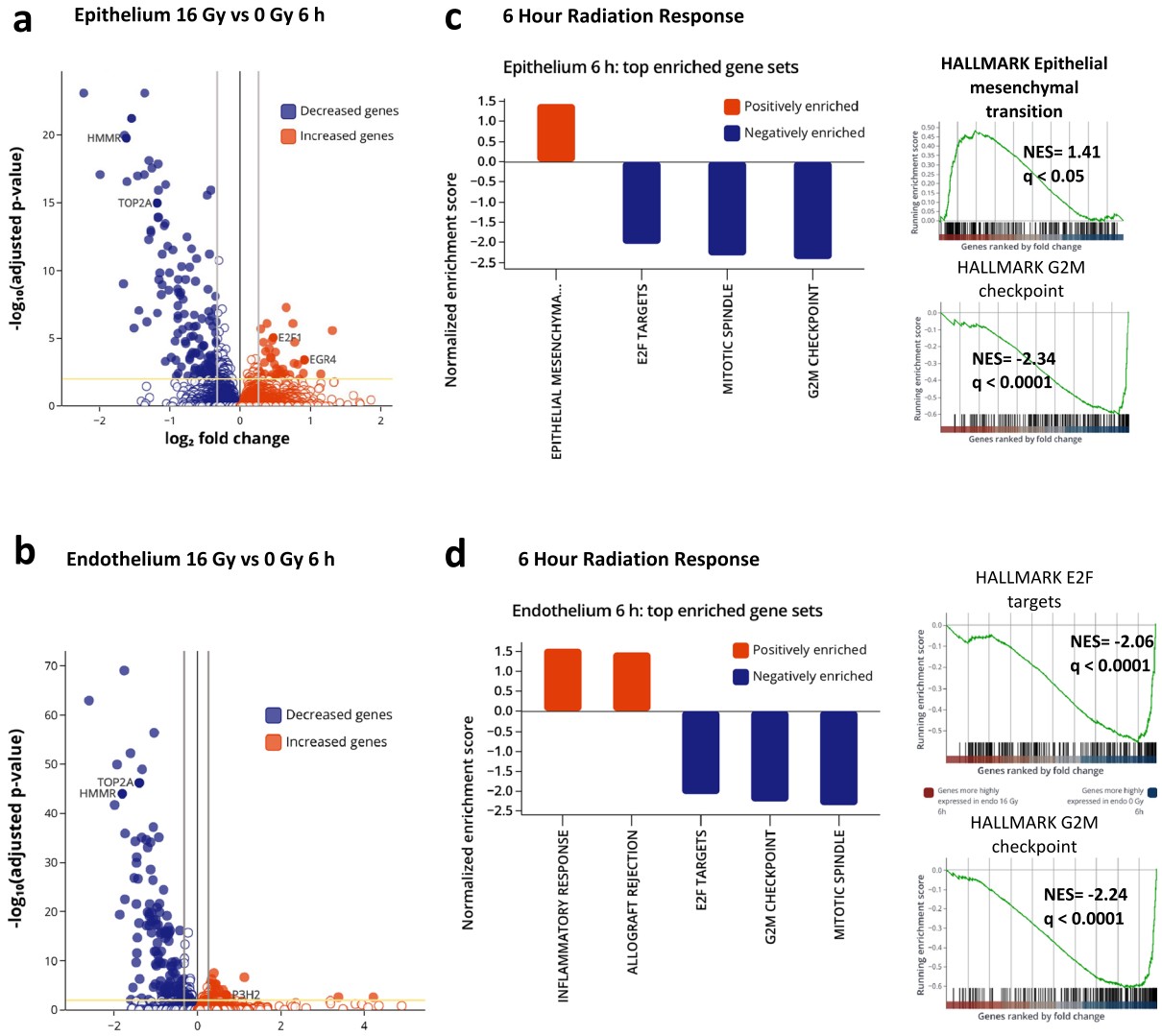

**Fig. 3 | Transcriptomic analyses shows that DNA damage and cell cycle arrest are early effects of radiation. a** Volcano plot of differentially expressed genes (DEGs) of alveolar epithelium, 6 h after radiation exposure, for adjusted $p < 0.01$. **b** Volcano plot of differentially expressed genes (DEGs) of endothelium, 6 h after radiation exposure, for adjusted $p < 0.01$. Adjusted $p$-values were obtained by applying Bonferroni correction for multiple comparison. **c** GSEA of Hallmark gene sets on the epithelium showing downregulation of hallmark E2F targets, mitotic spindle, and G2M checkpoints and upregulation of EMT, with $q < 0.05$. **d** GSEA of the endothelium showing downregulation of hallmark E2F targets, G2M checkpoints, and mitotic spindle, and upregulation of inflammatory responses. Plots created with Pluto (https://pluto.bio).

In addition, GO analysis revealed an increase in vasculature development pathways (Supplementary Fig. S4f), which is consistent with induction of a healing response to endothelial injury.

During lung injury, the removal of accumulated edema fluid in airspaces is facilitated by apical membrane epithelial Na$^+$ channels and hence, impairment of active Na$^+$ transport due to damage to the alveolo-capillary barrier can lead to reduced clearance of edema fluid from the alveolar space. Interestingly, the radiated Alveolus Chips showed significant downregulation of the sodium channel gene *SCNN1A* in the epithelium at 6 h, and subsequently, in the endothelium at 7d as well. GO analysis of the transcriptomics data also revealed that the endothelium upregulates several pathways related to calcium and metal ion transport, which may contribute to disruption of the alveolar epithelial-endothelial barrier as well.

### Hemoxygenase as a central regulator of RILI
Both transcriptomic analyses (Fig. 3a, b) and qPCR (Fig. 5a) revealed that *HMOX1*, which converts heme to biliverdin, Fe$^{2+}$, and CO as part of

an anti-oxidant response (Fig. 5b)[36], is upregulated in both the epithelium and the endothelium at 6 h post radiation. Interestingly, however, while *HMOX1* levels dropped to baseline in the epithelium by day 7, they remained high in the endothelium at this time (Fig. 5a). Moreover, an independent analysis of the transcriptomics data using a Network Model for Causality-Aware Discovery (NeMoCAD) algorithm[37,38] revealed that *HMOX1* and *IL6* were central targets in the top 10 (of 538) regulatory networks (Supplementary Fig. S4g). As the NeMoCAD algorithm has been previously used to predict therapeutic targets and repurpose drugs that reverse disease states[37,38], this suggests that the initial upregulation of *HMOX1* may provide radioprotection and that elevation of *HMOX1* could alleviate radiation injury in the lung. But recent studies have indicated that *HMOX1* may play a dual role in lung injury apart from its beneficial anti-oxidant and anti-inflammatory properties, as it can also promote cell death via ferroptosis[39]. At 7 d post-irradiation, several ferroptosis markers were affected in the endothelium, and both downregulation of *GPX4* and upregulation of *PTGS2* and *CHAC1* were observed

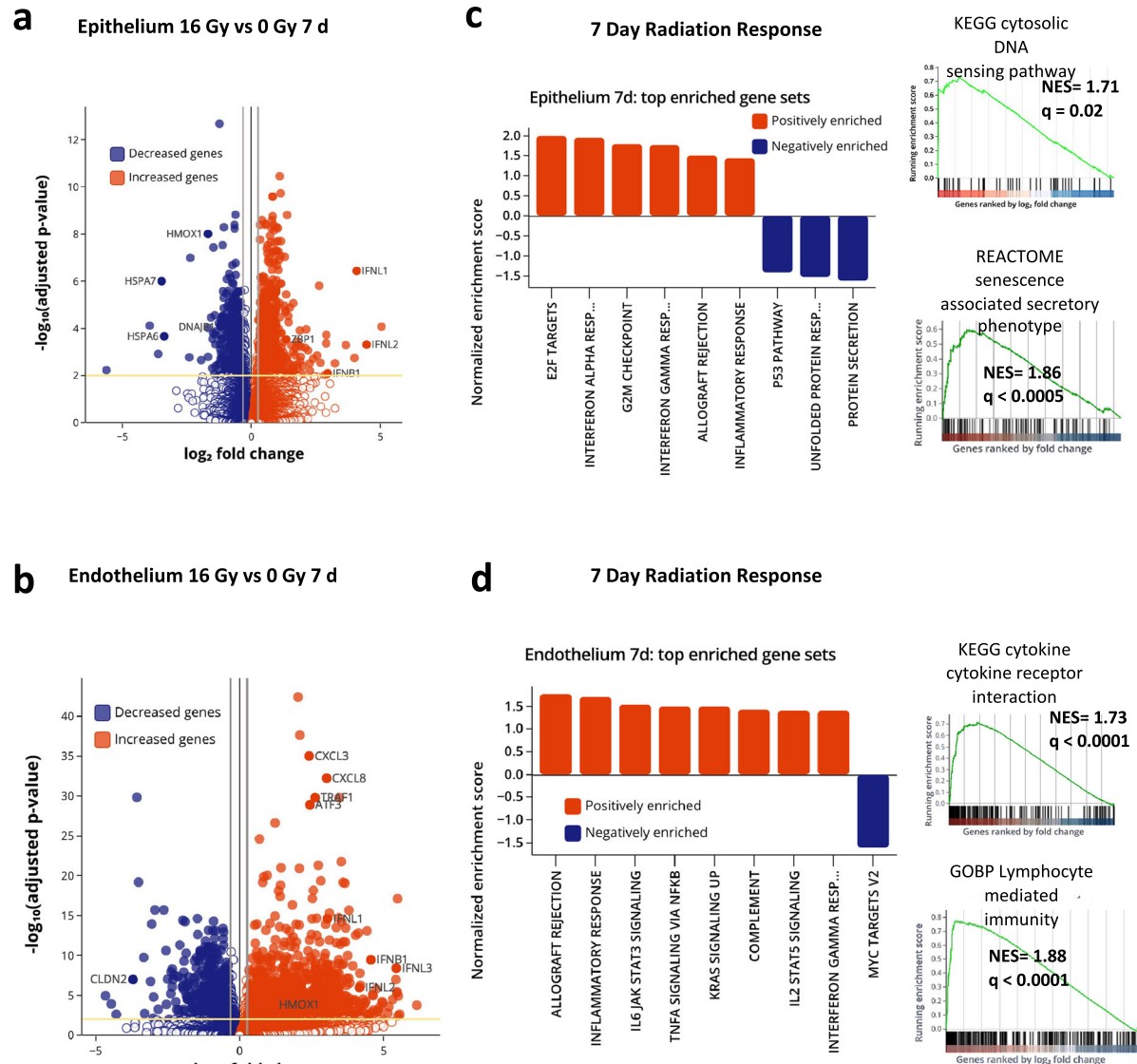

**Fig. 4 | Progressive inflammation and higher endothelial susceptibility to radiation-injury 7d post radiation. a** Volcano plot of differentially expressed genes (DEGs) of alveolar epithelium, 7 d after radiation exposure, for adjusted $p < 0.01$. **b** Volcano plot of differentially expressed genes (DEGs) of the endothelium 7 d after radiation exposure, for adjusted $p < 0.01$. Adjusted $p$-values were obtained by applying Bonferroni correction for multiple comparison. **c** GSEA of Hallmark gene sets of the epithelium showing upregulation of hallmark interferon response, cytosolic DNA sensing pathways and senescence associated secretory pathways, heatmaps showing top genes affected in the upregulation of cytosolic DNA sensing and chemokine. **d** GSEA of Hallmark gene sets on the endothelium showing upregulation of inflammatory response, TNF-a signaling, complement activation. GSEA was also performed for gene sets of Reactome, KEGG and GO: biological process and showed upregulation of the cytokine expression and NF-kB signaling. Representative enrichment plots showing increased cytokine-cytokine receptor interaction, lymphocyte mediated immunity are shown here. Plots created with Pluto (https://pluto.bio).

(Supplementary Fig. S5a), indicating that ferroptotic pathways may be active at this time[40,41].

## Investigating role of Hemoxygenase in RILI

To explore whether hemoxygenase plays an active role in treating RILI, we treated Lung Alveolus Chips with lovastatin (known upregulator of *HMOX1*[42]; 3 h prior to radiation exposure. Indeed, lovastatin-treated chips showed a significant increase in *HMOX1* and HO-1 expression (Fig. 5c, d) that was accompanied by a suppression of acute radiation injury and cellular DNA damage, as indicated by a lower number of 53bp1 nuclear foci in both the epithelium (Fig. 5e) and endothelium (Fig. 5f) at 6 h post-radiation (Supplementary Fig. S5b). Lovastatin also reduced epithelial cell hypertrophy (Fig. 5g and Supplementary Fig. S5c) and suppressed the inflammation response as indicated by a significant decrease in IL-6, IL-8, TNF-α,

and ICAM-1 levels when measured 1 day after radiation exposure (Fig. 5h). As a positive control, we prophylactically treated chips with the commonly used radiation countermeasure drug, prednisolone, prior to irradiation because it is commonly administered during radiotherapy to reduce inflammation. Indeed, prednisolone significantly reduced the number of 53bp1 foci both in the epithelium and endothelium (Fig. 5e, f). But while pretreatment with prednisolone also prevented cell hypertrophy (Supplementary Fig. S5c), reduced fluid accumulation in the air space (Supplementary Fig. S5d), and decreased the levels of some cytokines (e.g., IL-6 and ICAM-1) at 24 h, IL-8 and TNF-α levels were still significantly higher than in control non-irradiated samples (Fig. 5h).

Lovastatin treatment also reduced ROS levels compared to the no-drug radiation controls (Fig. 5i). Overall, the lovastatin treatment suppressed DNA damage and inflammation to a greater degree than

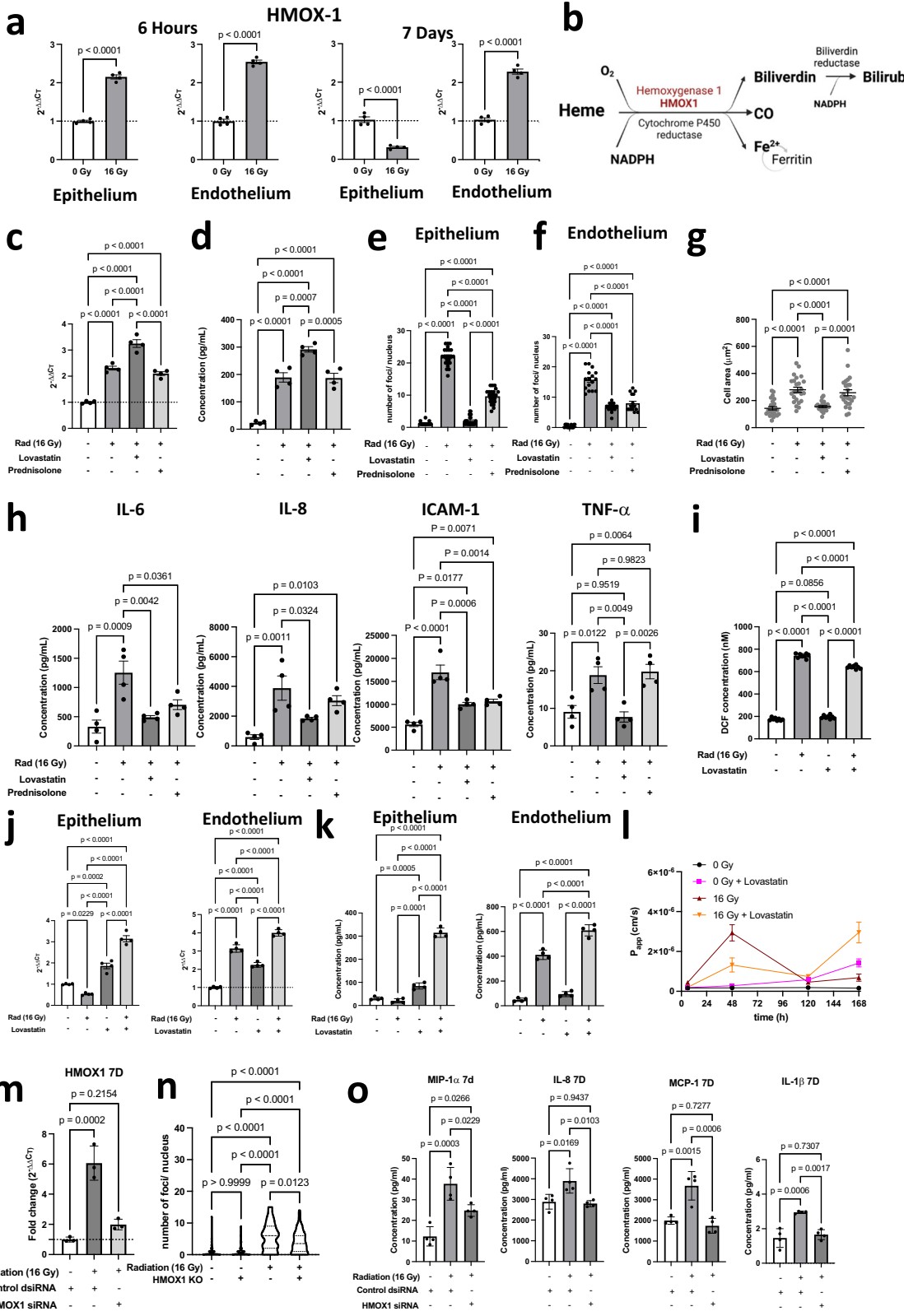

prednisolone when measured during the first day after radiation exposure. Unfortunately, in extended studies we found that while lovastatin initially protected against tissue barrier disruption for at least up to 48 h after radiation injury, permeability increased significantly after 5 d (Fig. 5i), and this was accompanied by visible fluid accumulation in the epithelial channel (Supplementary Fig. S5d). The presence of lovastatin elevated *HMOX1* levels at 7 d, both in the

endothelium and epithelium (Fig. 5j, k). Onset of barrier disruption in the presence of lovastatin was earlier than that in chips exposed to 16 Gy radiation in the absence of the drug (Fig. 5l), raising the possibility that prolonged upregulation of *HMOX1* maybe detrimental to cells.

To further explore the role of *HMOX1* in this response, we introduced siRNA directed against *HMOX1* through the endothelial channel

**Fig. 5 | Identification of HO-1 as a therapeutic target for RILI and testing the effect of lovastatin and prednisolone on RILI. a** Gene expression levels of *HMOX1* in the epithelium and endothelium at 6 h and 7d post radiation, Two-tailed Student's *t* test with Welch's correction $n = 4$ chips, F(epi 6 h) = 8.8, F(endo_6h) = 3.9, F(epi_7d) = 3.3, F(endo_7d) = 17.4, dF = 3, $p < 0.001$. **b** Pathway showing mechanism of action of *HMOX1* towards Bilirubin formation accompanied by the release of $Fe^{2+}$ and CO (created with BioRender.com). **c** Effect of lovastatin and prednisolone of *HMOX1* RNA level, 6 h after radiation exposure, one-way ANOVA, $n = 4$ chips, $F = 1.46$, DF = 3, $p < 0.05$ and (**d**) on HO-1 protein, 24 h after radiation exposure. One-way ANOVA, $n = 4$ chips, $F = 8.56$, DF = 3, $p < 0.05$. **e** Quantification of 53BP1 foci/nucleus in the epithelium and (**f**) endothelium. $n = 23$ nuclei analyzed in 3 chips, Ordinary one-way ANOVA, significance at $p < 0.0001$, F (epi) = 580.5, F(endo) = 122.3, dF = 3. **g** Effect of lovastatin and prednisolone on cellular hypertrophy in the alveolar epithelium, $n = 25$ cells for each condition, Ordinary one-way ANOVA with Tukey's test, significance at $p < 0.05$. **h** Effect of lovastatin and prednisolone on the cytokine response induced by radiation injury 24 h post-exposure. Representative panel showing the cytokines that were affected by the drug treatments. One-way ANOVA, $n = 4$ chips, DF = 3, F(IL-6) = 10.7, F(IL-8) = 10.3, F(ICAM-1) = 28.3 and F(TNF-a) = 12.0. **i** ROS assay 2 h after radiation exposure showing that presence of lovastatin decreases ROS levels in radiation exposed samples, $n = 8$ chips, one-way ANOVA, $F = 2677$, DF = 3, $p < 0.05$. **j** *HMOX1* expression, 7 d post-radiation by qPCR, $n = 4$ chips, one-way ANOVA, F(epi) 1.7, F(endo) = 282.3, DF = 3, $p < 0.0001$ and (**k**) HO-1 expression by ELISA assay, 7 post-radiation, indicating that *HMOX1* remains upregulated in both epithelium and endothelium in the presence of lovastatin, $n = 4$ chips, one-way ANOVA, F(epi) = 418.9, F(endo) = 277.3, DF = 3, $p < 0.0001$. **l** Changes in barrier permeability over time from 6 h to 7d post-radiation. **m** *HMOX1* expression by qPCR 7 d after treating the endothelium with control scrambled (*control dsiRNA*) or *HMOX1* siRNA, $n = 3$ chips, one-way ANOVA, $F = 45.0$, DF = 2, $p < 0.05$. **n** Number of g-H2AX foci per nucleus, showing that knockdown of HMOX1 shows lower number of foci, $n = 58$ nuclei, One-way ANOVA, $F = 43.7$, DF = 3, $p < 0.05$. **o** Cytokine profile shows that some inflammatory cytokines are downregulated post-*HMOX1* knockdown. One-way ANOVA, $n = 3$ chips, DF = 2, F(MIP-1a) = 21.0, F(IL-8) = 8.9, F(MCP-1) = 20.8 and F(IL-1b) = 20.1. All Data shown are mean +/− S.D.

---

to knock down HMOX1 expression. Knock down of *HMOX1* (Fig. 5m) resulted in a significant reduction in the number of (γ-H2AX) foci per nucleus post-*HMOX1* knockdown (Fig. 5n), confirming that HMOX1 normally helps to prevent DNA damage. Evaluation of the inflammation profile by cytokine analysis showed that the inflammatory markers, MIP-1α, IL-8, IL-1β and MCP-1 that were upregulated at day 7 post-exposure to radiation, were downregulated in samples in which HMOX1 was knocked down (Fig. 5o). However, other important inflammatory mediators, including IL-6, ICAM-1 and TNF-α, remained upregulated even after *HMOX1* knockdown, indicating that factors other than *HMOX1* are at play in acute RILI.

## Discussion

Recent guidelines of the American Thoracic Society[43] suggest that animal models of acute lung injury should mimic 3 out of 4 criteria: 1) histopathologic evidence of tissue injury, 2) alteration of the alveolar-capillary barrier, 3) an inflammatory response, and 4) physiological dysfunction. Our results show that human Lung Alveolus Chips that experience dynamic fluid flow and breathing motions can be used to recapitulate these hallmark features of RILI observed in human lung when exposed to a radiation dose of 16 Gy in vitro, whereas a static Transwell model that incorporates the same cells cultured under an ALI does not. Specifically, we found that exposure to this high dose of radiation induces DNA damage, cellular hypertrophy, tight junction disruption, increased barrier permeability, ROS production, and subsequent fluid accumulation on-chip.

DNA damage is often characterized by analyzing DSB number and morphology, and we observed a radiation dose-dependent response in DSB formation in both the epithelium and endothelium. This is consistent with the finding that as radiation dosage increases, the 53bp1 foci cluster into larger complexes that may represent repair centers[44]. We also observed induction of a senescence-associated secretory phenotype in response to radiation on-chip, which is another classic characteristic of radiation-pneumonitis in vivo[45]. Importantly, this Organ Chip model also replicates progressive inflammation, as demonstrated by production of clinically relevant early (IL-6, IL-1α, TNF-α etc.) and late (IL-1β, TGF-β, GM-CSF) stage cytokine markers of RILI, whereas this was not observed in a static Transwell culture. The finding that inflammation of the endothelium is also more prominent at day 7 after radiation exposure on-chip is interesting and is consistent with radiation-pneumonitis developing beginning ~1 week after exposure in radiated patients[3,5].

Cytosolic DNA sensing has been previously reported to mediate radiation-induced injury when analyzing in vivo models[34,46] and we observed upregulation of genes associated with this pathway in the Alveolus Chip. This has been shown to be mediated by upregulation of *ZBP1*[34], which is a Z-DNA-binding protein that may be induced by

interferons[35]; *ZBP1* also was upregulated in response to radiation in the Lung Chip model. In addition, we observed activation of inflammatory pathways, including TNF-α signaling via NF-κB, which was accompanied by *FCERI* mediated NF-kB signaling. Upregulation of *EDN1* and *OLR1* are also consistent with past studies that suggest endothelial apoptosis and ROS generation are upstream of radiation-induced epithelial cell injury and DNA damage responses[47,48].

Finally, transcriptomic analysis combined with use of a machine learning-based computational network analysis algorithm (NemoCAD)[37,38] suggested that *HMOX1* may play a central role in the RILI response, and hence, it could represent a potential therapeutic target. Additional experimental results confirmed that *HMOX1* expression is induced both in the epithelium and endothelium at 6 h post-radiation, which is consistent with past work demonstrating that it can be upregulated and provide a cytoprotective function in other lung diseases (e.g., interstitial pneumonia, malaria-associated lung injury, silicosis)[49–51]. Importantly, when we used lovastatin to artificially upregulate *HMOX1* levels in Lung Alveolus Chips exposed to radiation, we observed significant suppression of DNA damage and cellular hypertrophy as well as a decrease in inflammation at 24 h after exposure. However, lovastatin treatment aggravated barrier disruption over time as indicated by disruption of the alveolar-capillary interface and increased fluid accumulation in the epithelial air channel. But independent studies in which we used siRNA to knock down *HMOX1* gene expression prior to radiation exposure also led to lower DNA damage and suppression of some inflammatory cytokines, but not all. Thus, these data suggest that *HMOX1* also plays a protective role in the lung's initial response to radiation-induced injury but there are many other players involved in this response.

The primary goals of this research were (1) to recapitulate the effects of radiation induced lung injury following a radiological incident in a lung organ-on-chip model and (2) to identify novel medical countermeasures for military and civilian biodefence applications. There is a need to develop an organ-on-chip system for this context given clinical testing is not feasible. Animal models, which are used for assessing the efficacy and FDA clearance of radiation medical countermeasures, are often not clinically predictive as they do not recapitulate human toxicities. Currently there are no FDA approved medical countermeasures for radiation induced lung injury in the event of a nuclear accident. Hence, we explored radiation doses as high as 16 Gy administered over 20 min in this study because the higher range overlaps with the dose range that has been reported to induce acute RILI in humans[28]. The recapitulation of radiation induced lung injury from high dose exposures is also generalizable to the radiation toxicity observed in the lung with therapeutic radiation exposure because the radiation dosage rate we used (0.0133 Gy/s) is well within the conventional dosage range (≤0.03 Gy/s) for experimental models of

radiation injury[52]. While radiotherapy was commonly given in hyper-fractionated low doses (0.5–3 Gy) in the past, it is now possible to target small tumors with ablative (high dosage) radiation therapies that deliver higher radiation doses that are focused (e.g., stereotactically) at the tumor site, which results in lower toxicity. Preclinical animal models have also previously used single high radiation doses (-15–25 Gy) to induce pneumonitis and fibrosis[53,54]. More importantly, in humans, targeted radiotherapy regimes can range from 13.5 to 20 Gy delivered in each treatment and the one-time radiation dose of 16 Gy used in the current model falls, for example, within the hypo-fractionated dosage regime that is currently used for treatment of thoracic cancers[28,55].

The human Lung Alveolus Chip RILI model offers many advantages over existing in vitro models because it more faithfully mimics key hallmarks of this disease. Perhaps most importantly, it replicates the sustained inflammation response that is central to RILI. Notably, it also enables us to observe the effects of a single, high dose radiation on the lung, mimicking radiological disaster-related exposures that are otherwise difficult to recapitulate in human clinical trials. Further, transcriptomic analyses enable us to identify potential therapeutic targets and identify mechanisms of RILI. But it also must be noted that this model has certain limitations. For example, past studies have shown that neutrophils are important contributors to acute RILI and fibroblasts play a key role in longer term responses over many weeks to months after exposure to radiation, but they are missing in our model. However, due to the modular design of the chip, neutrophils, fibroblasts, and other potential cellular contributors can be integrated into the model in the future as this has been done in the past[25–27,56,57] to explore their contributions to both short and long term responses. Another limitation is that whole body radiation exposure induces injuries in multiple organs simultaneously (e.g., bone marrow, intestine) and there could be systemic cross talk that contributes to RILI from these distant sites via the circulation[58]. This is more easily studied in animal models; however, they do not faithfully mimic human organ radiation dose sensitivities or injury responses in the lung. Thus, future studies could be carried out in which the human Lung Alveolus Chip is linked fluidically with Organ Chip models via their vascular endothelium-lined channels as was done previously[58] to explore these systemic effects that are missing in the present study. Nevertheless, the results shown here demonstrate that many features of RILI observed in human patients can be replicated in vitro using the Alveolus Chip but not using more conventional static culture models. Thus, the Lung Alveolus Chip may represent a useful preclinical model for discovery of new radiation countermeasure drugs as well as for identification of potential biomarkers, mechanisms, and drug targets of RILI progression.

## Methods

### Lung Alveolus cultures in microfluidic chips
Two-channel microfluidic chips (Chip-S1) and automatic fluid handling ZOE systems were obtained from Emulate Inc (Boston, MA, USA). The two channels in the chip device are separated by a porous (8 μm pore size), PDMS membrane. After activating the culture surface of the chip using ER1/ER2 following manufacturer's instructions, the channels were coated with 200 μg/mL Collagen IV (5022-5MG, Advanced Biomatrix) and 15 μg/mL laminin (L4544-100 UL, Sigma) at 37 °C overnight in a $CO_2$ incubator (day 0). After coating is complete (day 1), chips are washed with DPBS ($+Ca^{2+}$, $+Mg^{2+}$), lung microvascular endothelial cells (Lonza CC-2527, P5) are perfused at a seeding density 8 ×$10^6$ cells/mL onto the bottom channel of the chips, inverted and allowed to attach to the underside of the membrane for 1 h. Following this, alveolar epithelial cells (Cell Biologics H-6053, P3) are seeded onto the top channel of the chip at a seeding density of 1.6 ×$10^6$ cells/mL and allowed to attach for 1 h. Following this, respective media are fed onto both the channels, and the chips are maintained under static

conditions overnight. On day 2, the chips are inserted into Pods (fluid feeding chambers, Emulate Inc.) and connected to the ZOE systems to ensure perfusion. The apical and basal channels are perfused with Alveolar epithelial growth medium (Cell Biologics, H6621) and endothelial growth medium (Lonza, EGM2-MV, CC-3202), respectively at a flow rate of 45 μL/h. On day 5, the apical medium is supplemented with 1 μM dexamethasone to enhance barrier formation. Following this, on day 7, medium is removed from the apical channel to establish an ALI. Chips continued to be perfused with EGM2-MV in the basal channel. On day 9, the perfusing medium was changed to EGM2-MV with 0.5% FBS. On day 11, the parameters on the ZOE instrument were changed to apply cyclic 5% mechanical strain, at 0.25 Hz frequency to mimic breathing motions of the lung. The mechanical strain is applied for 4 days before the chips are ready for treatment. This 14-day incubation ensures an alveolar-capillary interface with good barrier function[26]. On day 15, a modified EGM2-MV (with 50 nM hydrocortisone, instead of 550 nM in the original formulation) is perfused into the basal channel prior to exposure to radiation.

### Exposure to radiation on chip
PBMCs (StemCell, #70025.1) were labeled with CytoTox green and perfused into the bottom channel at a density of $10^6$ cells/mL for 5 min at a flow rate of 500 mL/h. Following PBMC perfusion, chips and attached Pods were removed from the ZOE and exposed to gamma radiation using a Cesium source (Cs-137). The chips were exposed to radiation ranging from 12 to 16 Gy (Fig. 1). Experiments shown in Figs. 2–5 include samples that were exposed to 16 Gy radiation. The sham irradiated control samples (0 Gy) were removed from the incubator and left at RT inside the biosafety cabinet. Following exposure to radiation, chips were connected back to the ZOE instrument and PBMCs were allowed to flow for 2 h at 500 μL/h. Following this, fresh medium was added to the bottom channel inlet and perfused at 45 μL/h.

### RNA extraction from alveolus Chip for qRT-PCR and RNA-seq
For RNA analyses, RNA was extracted at 6 h and 7 d post-radiation exposure to assess early and late effects of radiation, respectively. RNA isolation is performed using the RNeasy Plus Micro Kit (Qiagen 74034), following manufacturer's instructions, except few modifications made to suit the alveolus chip system. Briefly, 10 μL of β-mercaptoethanol was added to 1 mL Buffer RLT plus. To extract RNA, an empty 200 μL filtered barrier tip was inserted at the top channel outlet and washed with 100 μL DPBS ($+Ca^{2+}$, $+Mg^{2+}$) to collect apical washes. To collect epithelial cell lysates, two empty 200 μl filtered tips are placed at the bottom channel inlet and outlet. These two tips are blocked by placing the thumb on the outlet and the fore finger on the inlet. This is done to prevent the flow of lysis buffer to the bottom (endothelial) channel. An empty 200 μl filtered tip is placed at the top channel outlet, 100 μl RNase easy lysis buffer (Qiagen, #74034) is pipetted into the top channel inlet to lyse epithelial cells. The lysis buffer is pipetted back and forth 3 times by pressing and releasing the micropipette plunger. The lysate is then collected in a clean labeled 1.5 ml RNase free tube. The cell lysates are either directly processed for RNA isolation or stored at −80 °C for RNA-seq analysis, or qPCR. Endothelial cell lysates are collected similarly by blocking the top (epithelial) channel. RNA isolation was performed following manufacturer's instructions after this step.

### RT-qPCR
Total RNA was isolated using the RNeasy Micro Plus Kit (Qiagen 74034) for isolation from chips or RNeasy Mini Plus Kit (Qiagen 74106) for isolating RNA from transwells or well-plates. RNA concentrations were quantified using a nanodrop spectrophotometer (Thermo Scientific™ 912A1100). After quantification, 100–300 ng of RNA was used for cDNA synthesis. Reverse transcription was achieved using iScript cDNA

Synthesis kit (Bio-Rad 1708891). Following this, quantitative real time PCR (qRT-PCR) was conducted using TaqMan™ Fast Advanced Master Mix (Thermo Fisher 444455) and analyzed on Quantstudio 7 Flex Real-Time PCR system (Thermo Fisher 4485701), following manufacturer's instructions with UNG incubation hold at 50 °C for 2 min Polymerase activation, Hold at 95 °C for 2 min PCR (40 cycles), Denature at 95 °C for 1 min and anneal/extend at 60 °C for 20 s. Primer-specificity was confirmed by melting curve analysis. Relative RNA levels were quantified by the $\Delta\Delta C_t$ method[59] and normalized to endogenous controls of HPRT1 (for lung alveolar epithelium),and B2M or GAPDH (for microvascular endothelium). All Taqman probes were purchased from Thermo Fisher Scientific (Supplementary Table S9).

### Immunostaining and confocal microscopy
At the designated end-points, typically 6 h and 7d after radiation exposure (unless specified otherwise), cells were fixed with 1 % paraformaldehyde (PFA) in warm medium and incubated at RT for 5 min. Subsequently, they were fixed with 4% PFA (in PBS(+Mg/+Ca)) for an additional 30 min at RT. Followed by washing with PBS (−Mg/−Ca) and stored at 4 °C or processed immediately for imaging. For immunostaining, the cells were permeabilized with 0.1% Triton-X-100 in PBS for 10 min, and blocked with 5% donkey serum or 1% BSA/PBS for 1 h at RT. The cells are then incubated with primary antibodies at specific dilutions, overnight at 4 °C under shaking conditions. Fluorescent probe-conjugated secondary antibodies were incubated (in case of unconjugated primary antibodies) with the cells, followed by nuclear staining with Hoechst. The list of primary antibodies and their respective dilutions are listed in Supplementary Table S10. All immunofluorescence images are representative and have been captured in at least 2 separate experiments.

### Reactive oxygen species (ROS) assay
To test ROS levels in the alveolus chip in response to radiation, the OxiSelect™ In vitro ROS/RNS assay (catalog no. STA-347) was used, following manufacturer's instructions. Briefly the assay employs a quenched fluorescent probe dichlorodihydrofluorescin DiOxyQ (DCFH- DiOxyQ). The probe is primed by a quench removal agent and stabilized to the DCFH form. ROS and RNS can react with DCFH, and gets rapidly oxidized to the fluorescent 2',7'- dichlorodihydrofluorescein (DCF). Fluorescence intensity of DCF is directly proportional to the ROS/RNS levels in the sample. Standard curve of known DCF concentration vs fluorescence intensity is plotted and used to calculate DCF concentration from the sample effluents.

### Permeability assay
To test permeability of the epithelial-endothelial barrier of the chips, tracer dye molecule, FITC-dextran (40 kDa) was perfused to basal inlet reservoir of the pods, at a concentration of 100 µg/mL of each tracer. The apical inlet reservoir was filled with fresh medium. The flow rates of the apical and basal compartments were set to 120 µL/h. After 2 h of perfusion, effluents were collected from the apical and basal outlets and concentrations of the tracers quantified by fluorescence spectroscopy on a BioTek well-plate reader. Apparent permeability ($P_{app}$) calculations were based on chip manufacturer's (Emulate) instructions.

### Cytokine analysis
Effluents from the basal (vascular) outlets were collected 24 h and 7 d after radiation exposure, and were analyzed for a 13- panel of cytokines, and chemokines like IL-1α, IL-6, IL-8, IL-1β, GM-CSF, TNF-α, MCP-1, MIP-1α, IP-10, ICAM-1, P-selectin, Serpine1, using customized ProcartaPlex assay kits (Invitrogen).The analyte concentrations are evaluated using a Bio-Plex 3D suspension array system and analyzed for standard curve fitting and concentration calculations with Bio-Plex Manager software (Bio-Rad, v 6.0). TGF-β (Cat. no. DB100B) and HO-1 (Cat no. OKBB00836) were measured using ELISA kits (R & D Systems).

### Bulk RNA sequencing
RNA-seq was performed by Azenta Life Sciences using a RNA-seq package including polyA selection and sequencing using an Illumina HiSeq for 150 bp pair-ended reads. These sequence reads were trimmed using Trimmomatic v.0.36 to eliminate adapter sequences and nucleotides with poor quality. Subsequently, these trimmed reads were mapped onto the *Homo sapiens* GRCh38 reference genome using STAR aligner v.2.5.2b. Unique gene hit counts were computed using Counts from the Subread package v.1.5.2 followed by differential expression analysis. Significant DEGs were defined as $\log_2$ (fold change) ≥0.5 and $p < 0.01$ to adjust for false discovery. Differential expression analysis was performed by comparing the different groups. Genes were filtered to include genes with at least 3 reads in at least 20% of samples in any group. Differential expression analysis was then performed using the DESeq2 R package[60], which tests for differential expression based on a model using negative binomial distribution. $\log_2$(fold change) was calculated for each comparison. Thus, genes with a positive $\log_2$(fold change) value had increased expression, while genes with a negative $\log_2$(fold change) value had decreased expression in the radiated (16 Gy) samples. Volcano plots showing the $\log_2$(fold change) of each gene on the x-axis and the $-\log_{10}$(adjusted $p$-value) on the y-axis. Points are filled (in blue or red) if the gene's adjusted $p$-value is ≤0.01. The false discovery rate (FDR) method was applied for multiple testing correction[61]. FDR-adjusted $p$-values are shown on the y-axis of the volcano plot. An adjusted $p$-value of 0.01 was used as the threshold for statistical significance. Analysis and figures for transcriptomic analyses were generated using Pluto (https://pluto.bio).

### Gene set enrichment analysis (GSEA)
Gene set enrichment analysis (GSEA) was using the fgsea R package and the fgseaMultilevel() function. The $\log_2$(fold change) from the 16 Gy vs 0 Gy differential expression comparison was used to rank genes. Hallmarks gene set collection from the Molecular Signatures Database (MSigDB) was curated using the msigdbr R package. Similarly, for the GSEA analysis for GO biological processes, we used C5: Gene ontology gene sets from the MSigDB database using the msigdbr R package. In both cases, prior to running GSEA, the list of gene sets was filtered to include only gene sets with between 5 and 1000 genes. The Adj $p$-value is the false discovery rate (FDR)-adjusted $p$-value. Adjusted $p$ values were obtained using Bonferroni correction for multiple comparisons. The NES indicates the normalized enrichment score (NES) computed by GSEA, which represents the magnitude of enrichment as well as the direction. A positive NES indicates more enrichment in the 16 Gy radiated group while a negative NES indicates more enrichment in the control 0 Gy group. The enrichment plot shows the genes ranked by $\log_2$(fold change) along the x-axis with the vertical ticks representing the location of the genes in this gene set.

### NemoCAD analysis
NemoCAD is a machine-learning based computational tool to analyze transcriptomics signatures that can be targeted to reverse the global transcriptomic changes observed due to radiation exposure[37,38]. This approach is agnostic because it is run without an a priori defined drug/gene target or mechanism of action. NemoCAD uses previously computed drug-gene or gene-gene interaction probabilities and DEG signatures of the treated or diseased states and corresponding controls. The algorithm identifies target compounds that can revert a transcriptional signature in the diseased state to one observed in the healthy control.

### Statistics
All reported experiments were performed at least 3 times, with at least 4 technical repeats in each group. Data is represented as mean ± S.D.

unless specified in the figure legend. Graph plotting and statistics were performed on GraphPad Prism (Version 9.4.0). Statistical significance for 2 group comparisons were determined using two-tailed Student's *t* test, using Welch's correction for unpaired *t*-test. For multiple comparisons, one way ANOVA was used with Dunnett's multiple comparison test when compared to a single control group. All n, *p*-values and F, df values are mentioned in the respective figure or the associated legends.

### Reporting summary

Further information on research design is available in the Nature Portfolio Reporting Summary linked to this article.

## Data availability

All bulk RNA sequencing data have been uploaded on the Gene Expression Omnibus (GEO) database and made publicly available with accession codes GSE242706 and GSE242840 Additional data are available in the Supplementary information. The data were analyzed using available data packages mentioned in the manuscript and no new or custom codes were created to analyze the data. There are no restrictions on data availability and raw data has been made available. Source data are provided with this paper.

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

## Acknowledgements

This research was sponsored by funding from the US Food and Drug Administration grant 75F40119C10098, and the Wyss Institute for Biologically Inspired Engineering. The authors would like to thank Alican Ozkan for helping with imaging γ-H2AX nuclear foci and B. Lubamba and S. Gilpin for their helpful advice during the early phase of this project.

## Author contributions

Contribution: Q.D. participated in the ideation, design and performance of all experiments and analyzed the data working with S.H., E.J., and D.E.I, who also supervised the work. A.J. participated in the performance of experiments, data collection and discussions. A.W. acquired preliminary data that formed the basis of early experiments. Q.D., R.J.M. and A.W. acquired confocal microscopy images. Y.M and A.J. conducted the HMOX1 knockdown experiments. Q.D., S.H., E.J and D.E.I. prepared the manuscript with input from all authors.

## Competing interests

D.E.I. is a founder, board member, and chairs the SAB of Emulate Inc., in which he also holds equity. The remaining authors declare no competing interests.
