## [Peer Review File · Nature Communications]

A human lung alveolus-on-a-chip model of acute radiation-induced lung injuryREVIEWER COMMENTS

Reviewer #1 (Remarks to the Author):

This is an interesting study of high quality, focused on the acute injury of the lung epithelium and lung endothelium caused by a single exposure to gamma radiation. The radiation induced injury was modeled using the microfluidic organ-on-a-chip model the authors have previously developed, where human alveolar epithelium is interfaced with pulmonary endothelium. The authors recognize and discuss the limitations of this simple and elegant model containing only two of several cell types that can play a role in the observed phenomena and their underlying mechanisms.

The study is well designed, the experimental data are clean and clearly stated, and the results have relevance to understanding and countering the effects of clinical and accidental radiation.

CRITIQUE

1. The alveolus-on-chip model of human lung was exposed to gamma radiation of 12-16 Gray. While this total dose is in the range of radiation dosages used clinically for cancer patients (up to 60 Gray), it was apparently applied as a single acute exposure, while the patients receive radiation in a fractionated manner, i.e., divided into a number of smaller doses delivered typically 5 times per week over a period of one to two months. Presumably, the extent of cellular damage may be different for a single acute dose and fractionated regime, and the underlying mechanisms may also differ.

I suggest the authors to clarify the exact protocol used for the radiation exposure (duration, intensity), and rationalize the selection of one single high dose of radiation instead of mimicking a clinical regimen. Also, it would be helpful to add some discussion of the predictive power of the model for recapitulating the radiation damage seen in patients, and how the predictions may be affected by the differences in radiation exposure.

2. It was observed that, shortly after irradiation, the lung endothelium showed more DNA damage, cellular hypertrophy, upregulation of inflammatory cytokines, and loss of barrier function than lung epithelium. It would be of interest to compare these findings with clinical observations, which also revealed some of the reported effects (e.g., the reduced ability for vascularization after radiation).

3. Transcriptomic and gene network analyses identified increased expression of hemoxygenase-1 (HMOX-1, known to have a cytoprotective role) and suggested that this gene is serving as a central mediator of radiation injury of the lung. While data are comprehensive and well presented, the reader would benefit from a more detailed discussion of the progression and extent of cellular damage.

4. Pharmacological stimulation of HMOX-1 activity also significantly reduced acute radiation induced lung injury, and enhanced damage at later times. This finding should also be discussed in more depth.

5. A data formatting suggestion: it would be helpful to show data using the same scale on the Y axis, whenever possible (e.g., Figure 2 c & d; Figure 5 e&f, j, k), to make the differences easier to appreciate.

6. Please label all relevant Figure 5 panels with "endothelium" or "epithelium" to facilitate direct comparisons.

Reviewer #2 (Remarks to the Author):

This work employs a microfluidic lung-on-chip model, including alveolar epithelial cells, HMVECs, and cyclic stretching, to evaluate the effects of radiation on lung epithelial cells. This platform fills a need

for modeling cellular responses because rodent models do not model multiple aspects of pathophysiology, including low pneumonitis and minimal pulmonary fibrosis. Although not emphasized in the text, the platform also enables the ability to isolate the effects on and contributions and cross-talk of multiple cell types. Although this work demonstrates the value of the lung-on-chip platform for these types of studies (including comparing it to trans wells), it requires considerable revision prior to being ready for publication:

Major:

1. Fig. 3a,b – the number of detectable genes in each cell type should be clarified in the results section.
2. Fig. 3c,d – the gene set analysis is challenging to interpret due to unusual orange-blue colormap used for individual genes. It is further not clear what the 0-1 scale represents here. Additionally, while the running enrichment score is informative, it is not critical to see this in the main text. Instead, it would enhance interpretation to provide an unbiased summary of top significantly enriched gene sets (including those that are enriched in both 16Gy and 0 Gy) and to include the number of genes represented in each gene set. Additionally, a table of enrichment scores for all gene sets should be included in the supplement. The fonts of the gene labels are inconsistent. The methods should clarify exactly which databases were used for the gene sets used and how many there were. Methods should also clarify the method of p value computation.
3. Fig. S3 – the vertical label in panel (d) is cut off. Panel (f), the labels are not readable. Also, unclear if this was a GO analysis or a GSEA using GO annotations (given that normalized enrichment scores are reported).
4. Figs. 4 and S4: same comments as for Figs. 3 and S3.
5. It is unclear from the methods how RNA was separately isolated from epithelial vs endothelial cells.
6. The central issue with this work is that the title and conclusion of the manuscript are “Hemoxygenase-1 as a key mediator of acute radiation pneumonitis revealed in a human lung alveolus-on-a-chip”, which is largely unsupported by the work conducted. Although the role of HMOX-1 as a possible mediator of protection may be hinted at by the noted effects of lovastatin on both HMOX-1 and phenotypic improvements, these data are not causal. Lovastatin exerts numerous effects via the MAPK and PI3K/Akt/mTor pathways, possibly among others. Modulation of these pathways is certainly relevant to protection against radiation induced damage, but does not causally implicate HMOX-1. In order to assert the role of HMOX-1 as a causal regulator, it is essential to knockout HMOX-1 and show that the protective effect of lovastatin is ablated. Instead, it appears that HMOX-1 was highlighted here because of its known role in multiple lung diseases.
7. Related to the above comment, this paper does an excellent job of defining the utility of the long-on-chip model, but it is unclear that it reveals new mechanistic biology, as it suggests. It will be suitable for publication with conclusions more focused on the utility of the chip as a model system.

Minor

8. Post hoc test for ANOVA should be indicated in figure legends.
9. Fig. 2b – a label for the stain is needed.
10. Figure 1h – it is unclear if this is a protein (ELISA) assay, or transcript.
11. Throughout the figures, it is often difficult to discern which cell type or what assay was executed based on the figure and legend. Additional labeling would benefit readability.

Reviewer #3 (Remarks to the Author):

This article describes the development of an acute radiation-induced lung injury (RILI) model using the Lung Alveolus Chip platform exposed to gamma radiation. Within 6 hours of radiation exposure, both the lung epithelium and endothelium showed evidence of DNA damage, cellular hypertrophy, upregulation of inflammatory cytokines, and loss of barrier function. The study also investigated the protective effects of prednisolone and found that increased expression of the cytoprotective gene hemoxygenase-1 (HMOX-1) was a central mediator of radiation-induced injury based on

transcriptomic analysis. While the manuscript is well-crafted, the results are not clear, but are ambiguous. The statement in the title "Hemoxygenase-1 as a key mediator of acute radiation pneumonitis" appears to be deceptive, in my opinion. Due to lack of novelty, originality, and noteworthy technical advancements in the chip system along with significant limitations in experimental conditions, it is not recommended for publication in this journal, at least in its current form. Specific comments are listed below:

1. Fibroblasts have key role in the progression of RILI, along with pathophysiological changes such as immune cell infiltration and cytokine release. Fibroblast differentiation as well as the synthesis and deposition of extracellular matrix (ECM) such as collagen are significant hallmarks of RILI. Therefore, to effectively study RILI, it is imperative to incorporate fibroblasts into the Lung Alveolus Chip and analyze the differentiation of fibroblasts as well as the deposition of ECM at various radiation levels.
2. Two types of alveolar epithelial cells, AT1 and AT2, cover the lung alveoli. AT1 cells account for 90-95% of the alveolar surface, while AT2 cells cover only 7%, yet AT2 cells are essential for epithelial repair. The ratio of AT1 and AT2 cells should also be measured at various radiation levels, and it is important to evaluate how this ratio changes with different radiation doses.
3. Limiting the accumulation of immune cells, such as neutrophils and monocytes that mainly differentiate into macrophages, has been shown to attenuate the pathogenesis of RILI. What is the relationship between the infiltration of immune cells at different radiation doses and the development of RILI? Which immune cells are primarily involved in the profibrotic process during radiation exposure?
4. Moreover, radiation-induced tissue hypoxia in the lung can trigger vascular endothelial mesenchymal transition (EndMT). The authors should explore the potential effects of different radiation doses on vascular endothelial mesenchymal transition (EndMT), in addition to epithelial to mesenchymal transition (EMT), to gain a better understanding of the permeability changes that result in pulmonary edema on the Lung Alveolus Chip.
5. Figure 5I depicts the impact of lovastatin on barrier permeability solely in the alveolar epithelium from 6 h to 7 days post-radiation. What could be the possible reasons for the initial decrease in barrier permeability in the presence of lovastatin around 120 hours, and the subsequent increase?
6. What was the impact of prednisolone on barrier permeability throughout the study period?
7. During lung injury, the removal of accumulated edema fluid in airspaces is facilitated by apical membrane epithelial Na⁺ channels. However, impairment of active Na⁺ transport due to damage to the alveolo-capillary barrier can lead to reduced clearance of edema fluid from the alveolar spaces. Therefore, it would be important for the authors to examine the impact of RILI on membrane epithelial Na⁺ channels.
8. It has been reported that prednisolone can reduce capillary permeability and increase the clearance of alveolar edema fluid. Did the accumulation of fluid differ between the absence and presence of prednisolone? Additionally, is it feasible to measure the clearance of fluid accumulated in the epithelial channel?
9. Furthermore, acute radiation pneumonitis is reversible which will resolve if further exposure is avoided. But in Lung Alveolus Chip system, it seems to be irreversible even under the treatment with protective drugs, prednisolone and lovastatin prior to irradiation. Treatment with these drugs showed higher levels of HMOX-1 expression at both the gene and protein levels, and this was accompanied by a lower number of 53bp1 nuclear foci in both the epithelium and endothelium at 6 h post-radiation. The protective effects of these drugs were observed at 6 h post-radiation which is too early to conclude that the treatment with these drugs and the increased HMOX-1 expression mediates the suppression of acute radiation injury. In addition, the permeability results in the presence of lovastatin contradict these results.
10. The tissue hypoxia in the lung following irradiation is also a contributing factor to RILI. Can authors provide information on the oxygen levels observed in the Lung Alveolus Chip at various radiation doses?
11. Literature suggests that early acute radiation pneumonitis typically develops 2-12 weeks after radiotherapy. Consequently, the experimental timeframe of 6 hours to 7 days may not be adequate to fully evaluate the consequences of radiation exposure.

12. What concentrations of prednisolone and lovastatin were used in the study?
13. What was the length of time that the chips were exposed to radiation within the range of 12-16 Gy during each exposure?
14. The legend of Figure 1D states that the quantification of 53BP1 foci was done per cell, but the figure shows the number of foci per nucleus. What methodology was utilized to determine the number of 53BP1 foci per nucleus/cell?

RESPONSE TO REVIEWERS

Reviewer 1:

1. The alveolus-on-chip model of human lung was exposed to gamma radiation of 12-16 Gray. While this total dose is in the range of radiation dosages used clinically for cancer patients (up to 60 Gray), it was apparently applied as a single acute exposure, while the patients receive radiation in a fractionated manner, i.e., divided into a number of smaller doses delivered typically 5 times per week over a period of one to two months. Presumably, the extent of cellular damage may be different for a single acute dose and fractionated regime, and the underlying mechanisms may also differ.

I suggest the authors to clarify the exact protocol used for the radiation exposure (duration, intensity), and rationalize the selection of one single high dose of radiation instead of mimicking a clinical regimen. Also, it would be helpful to add some discussion of the predictive power of the model for recapitulating the radiation damage seen in patients, and how the predictions may be affected by the differences in radiation exposure.

We realize that we did not clearly explain the primary goals of our study. This project was funded by the FDA through a U.S. Congressional Act focused on developing responses to potential nuclear threats. Thus, we were asked to explore whether the human lung-on-a-chip (Lung Chip) technology could be used to recapitulate the effects of acute radiation induced lung injury following a radiological incident or nuclear disaster. A secondary goal was to explore whether this model could be used to identify novel medical countermeasures (MCMs) for acute radiation injury for military and civilian nuclear defense applications. Our project was funded because the FDA recognized that there is a great need for human-relevant preclinical models for this purpose as clinical testing in humans is not feasible and animal models, which are currently relied upon for assessment of radiation MCMs by the FDA, are often not predictive of human responses. Moreover, there currently are no FDA approved MCMs for radiation-induced lung injury in the event of a nuclear accident. Hence, we explored radiation doses as high as 16 Gy administered over 20 min in this study because the higher range overlaps with the dose range that has been reported to induce acute RILI in humans [1-3; References are listed at the end of this document].

Importantly, the recapitulation of radiation induced lung injury from high dose exposures is also generalizable to the radiation toxicity observed in the lung with therapeutic radiation exposure because the radiation dosage rate we used (0.0133 Gy/s) is well within the conventional dosage range (≤ 0.03 Gy/s) for experimental models of radiation injury [4]. While radiotherapy was commonly given in hyperfractionated low doses (0.5 -3 Gy) in the past, it is now possible to precisely target small tumors with ablative (high dosage) radiation therapies that deliver higher radiation doses that are focused (e.g., stereotactically) at the tumor site, which results in lower toxicity. Preclinical animal models have also previously used single high radiation doses (~15-25 Gy) to induce pneumonitis and fibrosis [5, 6]. More importantly, in humans, targeted radiotherapy regimes can range from 13.5-20 Gy delivered in each treatment and the one-time radiation dose of 16 Gy used in the current model falls, for example, within the hypofractionated dosage regime that is currently used for treatment of thoracic cancers[7, 8]. We now discuss these points in the Introduction and Discussion.

2. It was observed that, shortly after irradiation, the lung endothelium showed more DNA damage, cellular hypertrophy, upregulation of inflammatory cytokines, and loss of barrier function than lung epithelium. It would be of interest to compare these findings with

clinical observations, which also revealed some of the reported effects (e.g., the reduced ability for vascularization after radiation).

Extensive clinical and molecular characterization of acute radiation-induced lung injury due to nuclear accident remains limited and clinical observations of RILI in the context of radiotherapy are mainly symptomatic and based on radiographic imaging. But we clearly note that the radiation dose sensitivity observed in the Lung Alveolus Chip more closely matched that observed in human lung than in animal preclinical models and that the hypertrophy of the alveolar epithelial cells we observed at 6 hours post radiation exposure (**Fig. 2b,c**) mimics a feature commonly seen in alveolar injury in vivo [1, 9]. We also observed induction of a senescence-associated secretory phenotype in response to radiation on-chip, which is another classic characteristic of radiation-pneumonitis in human lung [10]. In addition, cytosolic DNA sensing has been previously reported to mediate radiation-induced lung injury when analyzing in vivo models [11, 12] and we also observed up regulation of genes associated with this pathway in the Alveolus Chip. Moreover, both our experiments with PBMCs and gene ontology analysis of the transcriptomic data shows that pathways related to neutrophil chemotaxis and migration, humoral and innate immune response are upregulated in response to radiation (Fig. S4f); all of these features have similarly been observed clinically [1, 13]. In addition, we now point out that our transcriptomic and immunostaining data also revealed evidence of endothelial-mesenchymal transition that also has been previously described in models of radiation-induced injury [14]. Importantly, a static Transwell culture model that incorporates a similar tissue-tissue interface and contact with air does not replicate these clinical features, thus demonstrating the novelty and potential value of this new Organ Chip model of acute RILI for the field.

3. Transcriptomic and gene network analyses identified increased expression of hemoxygenase-1 (HMOX-1, known to have a cytoprotective role) and suggested that this gene is serving as a central mediator of radiation injury of the lung. While data are comprehensive and well presented, the reader would benefit from a more detailed discussion of the progression and extent of cellular damage.

&

4. Pharmacological stimulation of HMOX-1 activity also significantly reduced acute radiation induced lung injury, and enhanced damage at later times. This finding should also be discussed in more depth.

We agree that we cannot make specific claims about the role of HMOX-1 in the radiation response in this study that was focused on developing a new human-relevant preclinical model of acute radiation injury in the lung. Thus, we have revised the Title, Abstract, Results, and Discussion to downplay this point. Nevertheless, we do describe the intriguing findings showing that there is an initial upregulation of HMOX-1 that correlates with radioprotection, which suggests that it would be important to explore whether this might represent a potential therapeutic target in future studies. This is complicated, however, as recent studies have indicated that HMOX-1 may play a dual role in lung injury and apart from its beneficial anti-oxidant and anti-inflammatory properties, as it also can promote cell death via ferroptosis. Interestingly, we found that despite the upregulation of HMOX-1 at 6 h post radiation, ferroptosis markers were not differentially expressed in radiated samples (not shown). However, at 7 d post-irradiation, several ferroptosis markers were affected in the endothelium, and both downregulation of GPX4 and upregulation of PTGS2 and CHAC1 were observed indicating that ferroptotic pathways are active at this time (new **Supplementary Fig. S5a**). We also analyzed the viability of the cells at 5 d post-radiation and observed that there was significant reduction in cell death viability in both the epithelium and endothelium in response to radiation (**Supplementary Fig. S1b**). In addition, we carried out new experiments and have added new

data showing that siRNA knock down of HMOX1 prevents some radiation induced lung responses, but not others (**Fig. 5m-o**).

5. A data formatting suggestion: it would be helpful to show data using the same scale on the Y axis, whenever possible (e.g., Figure 2 c & d; Figure 5 e&f, j, k), to make the differences easier to appreciate.

We appreciate the suggestion and have made these changes to all the suggested figures except for **Fig. 2d** because the major point that we are trying to visualize are the presence or absence of significant differences between the radiated and unirradiated condition within each cell type, and this would become much less clear if we used the same scale. Thus, we have chosen the leave **Fig. 2d** as is.

6. Please label all relevant Figure 5 panels with “endothelium” or “epithelium” to facilitate direct comparisons.

We have made these changes in the manuscript, as requested.

Reviewer 2:

1. Fig. 3a,b – the number of detectable genes in each cell type should be clarified in the results section.

We now clarify that 11976 were genes detected in the epithelium and 19661 genes in the endothelium.

2. Fig. 3c,d – the gene set analysis is challenging to interpret due to unusual orange-blue colormap used for individual genes. It is further not clear what the 0-1 scale represents here. Additionally, while the running enrichment score is informative, it is not critical to see this in the main text. Instead, it would enhance interpretation to provide an unbiased summary of top significantly enriched gene sets (including those that are enriched in both 16Gy and 0 Gy) and to include the number of genes represented in each gene set. Additionally, a table of enrichment scores for all gene sets should be included in the supplement. The fonts of the gene labels are inconsistent. The methods should clarify exactly which databases were used for the gene sets used and how many there were. Methods should also clarify the method of p value computation.

We have made these modifications to the Figures, as requested. We have also added additional tables to the Supplementary Information (Table S1-S8), with all the requested details.

3. Fig. S3 – the vertical label in panel (d) is cut off. Panel (f), the labels are not readable. Also, unclear if this was a GO analysis or a GSEA using GO annotations (given that normalized enrichment scores are reported).

We have made the requested corrections. This is a GSEA analysis using GO annotations, which we now explain more clearly in the legend and the Methods section.

4. Figs. 4 and S4: same comments as for Figs. 3 and S3.

We have made the requested corrections. This is a GSEA analysis using GO annotations, which we now explain more clearly in the legend and the Methods section.

5. It is unclear from the methods how RNA was separately isolated from epithelial vs endothelial cells.

Epithelial cell lysates for qPCR or RNA-seq are collected by lysis buffer through the epithelial channel while physically occluding flow to the endothelial channel, which prevents entry of the lysis buffer. Endothelial cell lysates are then collected similarly by physically blocking the epithelial channel while passing lysis buffer through the endothelial channel. A detailed description of the method has been added to the Methods section.

6. The central issue with this work is that the title and conclusion of the manuscript are “Hemoxygenase-1 as a key mediator of acute radiation pneumonitis revealed in a human lung alveolus-on-a-chip”, which is largely unsupported by the work conducted. Although the role of HMOX-1 as a possible mediator of protection may be hinted at by the noted effects of lovastatin on both HMOX-1 and phenotypic improvements, these data are not causal. Lovastatin exerts numerous effects via the MAPK and PI3K/Akt/mTor pathways, possibly among others. Modulation of these pathways is certainly relevant to protection against radiation induced damage, but does not causally implicate HMOX-1. In order to assert the role of HMOX-1 as a causal regulator, it is essential to knockout HMOX-1 and show that the protective effect of lovastatin is ablated. Instead, it appears that HMOX-1 was highlighted here because of its known role in multiple lung diseases.

We agree that we cannot make specific claims about the role of HMOX-1 in the radiation response in this study and that we did not clearly communicate that this manuscript is focused on developing a new human-relevant preclinical model of acute radiation injury in the lung. Thus, we have revised the Title, Abstract, Results, and Discussion to downplay this point. Nevertheless, we do describe the intriguing findings showing that there is an initial upregulation of HMOX-1 that correlates with radioprotection, which suggests that it would be important to explore whether this might represent a potential therapeutic target in future studies. This is complicated, however, as recent studies have indicated that HMOX-1 may play a dual role in lung injury and apart from its beneficial anti-oxidant and anti-inflammatory properties, as it also can promote cell death via ferroptosis. Interestingly, we found that despite the upregulation of HMOX-1 at 6 h post radiation, ferroptosis markers were not differentially expressed in radiated samples (not shown). However, at 7 d post-irradiation, several ferroptosis markers were affected in the endothelium, and both downregulation of GPX4 and upregulation of PTGS2 and CHAC1 were observed indicating that ferroptotic pathways are active at this time (new **Supplementary Fig. S5a**). We also analyzed the viability of the cells at 5 d post-radiation and observed that there was significant reduction in cell death viability in both the epithelium and endothelium in response to radiation (**Supplementary Fig. S1b**). In addition, we carried out new experiments and have added new data showing that siRNA knock down of HMOX1 prevents some radiation induced lung responses, but not others (**Fig. 5m-o**).

7. Related to the above comment, this paper does an excellent job of defining the utility of the long-on-chip model, but it is unclear that it reveals new mechanistic biology, as it suggests. It will be suitable for publication with conclusions more focused on the utility of the chip as a model system.

We agree with the reviewer and have made modifications to the Title and all parts of the manuscript to clarify that the focus of this work is on the development and validation of the utility of a new human-relevant preclinical model of acute radiation injury in the lung.

Minor-

8. Post hoc test for ANOVA should be indicated in figure legends. - Done

9. Fig. 2b – a label for the stain is needed. - Done

10. Figure 1h – it is unclear if this is a protein (ELISA) assay, or transcript. -This has been clarified in the text.

11. Throughout the figures, it is often difficult to discern which cell type or what assay was executed based on the figure and legend. Additional labeling would benefit readability. - Requested changes have been made in the revised manuscript.

Reviewer 3:

General-

The statement in the title "Hemoxygenase-1 as a key mediator of acute radiation pneumonitis" appears to be deceptive, in my opinion.

We agree that we cannot make specific claims about the role of HMOX-1 in the radiation response in this study that was focused on developing a new human-relevant preclinical model of acute radiation injury in the lung. Thus, we have revised the Title, Abstract, Results, and Discussion to downplay this point. Nevertheless, we do describe the intriguing findings showing that there is an initial upregulation of HMOX-1 that correlates with radioprotection, which suggests that it would be important to explore whether this might represent a potential therapeutic target in future studies. This is complicated, however, as recent studies have indicated that HMOX-1 may play a dual role in lung injury and apart from its beneficial anti-oxidant and anti-inflammatory properties, as it also can promote cell death via ferroptosis. Interestingly, we found that despite the upregulation of HMOX-1 at 6 h post radiation, ferroptosis markers were not differentially expressed in radiated samples (not shown). However, at 7 d post-irradiation, several ferroptosis markers were affected in the endothelium, and both downregulation of GPX4 and upregulation of PTGS2 and CHAC1 were observed indicating that ferroptotic pathways are active at this time (new **Supplementary Fig. S5a**). We also analyzed the viability of the cells at 5 d post-radiation and observed that there was significant reduction in cell death viability in both the epithelium and endothelium in response to radiation (**Supplementary Fig. S1b**). In addition, we carried out new experiments and have added new data showing that siRNA knock down of HMOX1 prevents some radiation induced lung responses, but not others (**Fig. 5m-o**).

1. Fibroblasts have key role in the progression of RILI, along with pathophysiological changes such as immune cell infiltration and cytokine release. Fibroblast differentiation as well as the synthesis and deposition of extracellular matrix (ECM) such as collagen are significant hallmarks of RILI. Therefore, to effectively study RILI, it is imperative to incorporate fibroblasts into the Lung Alveolus Chip and analyze the differentiation of fibroblasts as well as the deposition of ECM at various radiation levels.

While fibrosis is an important component of the long-term response to radiation injury that develops over many weeks to months, it is not a central feature of the focus of the present study that specifically attempts to model the *acute* response of lung to radiation injury. We now more clearly explain this in the Introduction and clarify that analysis of radiation-induced fibrosis

is beyond the scope of this study. In the Discussion, we now clarify that *"past studies have shown that neutrophils are important contributors to acute RILI and fibroblasts play a key role in longer term responses over many weeks to months after exposure to radiation, but they are missing in our model. However, due to the modular design of the chip, neutrophils, fibroblasts, and other potential cellular contributors can be integrated into the model in the future as this has been done in the past to explore their contributions to both short and long term responses."*

2. Two types of alveolar epithelial cells, AT1 and AT2, cover the lung alveoli. AT1 cells account for 90-95% of the alveolar surface, while AT2 cells cover only 7%, yet AT2 cells are essential for epithelial repair. The ratio of AT1 and AT2 cells should also be measured at various radiation levels, and it is important to evaluate how this ratio changes with different radiation doses.

As requested, we analyzed effects on alveolar Type I and II cells by quantifying expression of biomarkers for each - aquaporin 5 (AQP5) and surfactant protein B (SPB), respectively. We detected a significant decrease in the expression of the alveolar Type I marker with a concomitant increase in the Type II marker at 6 hr after exposure to 16 Gy radiation (**Supplementary Fig. S1g**), although there was no significant change in expression at lower doses. This finding among others led us to focus the bulk of our studies using the 16 Gy radiation dosage.

3. Limiting the accumulation of immune cells, such as neutrophils and monocytes that mainly differentiate into macrophages, has been shown to attenuate the pathogenesis of RILI. What is the relationship between the infiltration of immune cells at different radiation doses and the development of RILI? Which immune cells are primarily involved in the profibrotic process during radiation exposure?

We demonstrated that radiation exposure of 16 Gy results in a large increase in recruitment of PBMCs that include monocytes and lymphocytes after 6 hrs and that this increases even further over 7 days in our study (**Fig. 2d**). We did not do a dose response because this study was focused on lung injury induced by radiation accident or nuclear disaster, as we now explain in the revised Introduction. While the question of which immune cells are primarily involved in the profibrotic response is a good one, pursuing this is beyond the scope of the present study that focuses on development of a human preclinical model of acute radiation-induced lung injury using organ chip technology.

4. Moreover, radiation-induced tissue hypoxia in the lung can trigger vascular endothelial mesenchymal transition (EndoMT). The authors should explore the potential effects of different radiation doses on vascular endothelial mesenchymal transition (EndMT), in addition to epithelial to mesenchymal transition (EMT), to gain a better understanding of the permeability changes that result in pulmonary edema on the Lung Alveolus Chip.

This is an excellent point and we now point out that our transcriptomic data revealed that EDN1, RUNX3, TGFB1, and CXCL8 genes are all upregulated in response to 16 Gy radiation (**Fig. 3a,b**), while there was a decrease in PECAM1 expression and loss of endothelial cell-cell junctions (**Supplementary Fig. S1d**). This is consistent with the induction of an endothelial-mesenchymal transition that has been previously described in models of radiation-induced injury, which causes endothelial cells to lose their polarity and become more migratory and invasive into surrounding tissue.

5. Figure 5l depicts the impact of lovastatin on barrier permeability solely in the alveolar epithelium from 6 h to 7 days post-radiation. What could be the possible reasons for the initial decrease in barrier permeability in the presence of lovastatin around 120 hours, and the subsequent increase?

While dissecting the mechanism that underlies this response is far beyond the scope of this study, our transcriptomics data suggest that there is an initial increase in DNA repair. Owing to the innate cellular radioprotection that may be mediated by HMOX1, the cells get into a DNA repair phase which is supported by our transcriptomics data, which shows that this may be because initially DNA repair, which may be regulated by cytoprotective genes, such as HMOX1. These repair mechanisms may lead to a decrease in the permeability at 120 h. However, prolonged upregulation of ROS levels, inflammation and HMOX1 upregulation may surpass the anti-oxidant activity of HMOX1 resulting in excessive Fe²⁺ accumulation and ferroptosis as we observed cell death 5 days after exposure (**new Suppl. Fig. S1b**). We also have added new data showing that siRNA knock down of HMOX1 prevents some radiation induced lung responses, but not others (**Fig. 5m-o**).

6. What was the impact of prednisolone on barrier permeability throughout the study period?

Prednisolone did not significantly alter barrier permeability (P_{app} values) compared to untreated control. However, prednisolone significantly reduced fluid accumulation when measured 5 days after radiation exposure as only 33% exhibited barrier disruption compared to 83% in the absence of drug treatment. These data have now been added to the manuscript (**new Suppl. Fig. S5d**).

7. During lung injury, the removal of accumulated edema fluid in airspaces is facilitated by apical membrane epithelial Na⁺ channels. However, impairment of active Na⁺ transport due to damage to the alveolo-capillary barrier can lead to reduced clearance of edema fluid from the alveolar spaces. Therefore, it would be important for the authors to examine the impact of RLI on membrane epithelial Na⁺ channels.

This is an excellent point. While carrying out direct Na⁺ transport measurements is beyond the scope of this study, the radiated samples show significant downregulation of the sodium channel gene SCNN1A in the epithelium at 6 h, and subsequently, in the endothelium at 7d as well. Further, gene ontology analysis of the transcriptomics data reveal that the endothelium also upregulates several pathways related to calcium and metal ion transport, that may contribute to disruption of the alveolar epithelial-endothelial barrier as well. We now explain this in the Results.

8. It has been reported that prednisolone can reduce capillary permeability and increase the clearance of alveolar edema fluid. Did the accumulation of fluid differ between the absence and presence of prednisolone? Additionally, is it feasible to measure the clearance of fluid accumulated in the epithelial channel?

We quantified the % of chips that displayed visually detectable fluid accumulation, and as the Reviewer suggested, we detected a significant reduction in fluid accumulation in radiated chips treated with prednisolone (now shown in new **Supplementary Fig. 5d**).

9. Furthermore, acute radiation pneumonitis is reversible which will resolve if further

exposure is avoided. But in Lung Alveolus Chip system, it seems to be irreversible even under the treatment with protective drugs, prednisolone and lovastatin prior to irradiation. Treatment with these drugs showed higher levels of HMOX-1 expression at both the gene and protein levels, and this was accompanied by a lower number of 53bp1 nuclear foci in both the epithelium and endothelium at 6 h post-radiation. The protective effects of these drugs were observed at 6 h post-radiation which is too early to conclude that the treatment with these drugs and the increased HMOX-1 expression mediates the suppression of acute radiation injury. In addition, the permeability results in the presence of lovastatin contradict these results.

Acute radiation pneumonitis may be reversible in animal models, however, this is not the case in humans, where pneumonitis often leads to pulmonary fibrosis. One of the primary merits of this human Lung Alveolus Chip model is that it allows us to capture early effects as well as progressive inflammation, as observed clinically. Drugs, such as prednisolone and lovastatin, may initially inhibit DNA damage and hypertrophy by upregulating cytoprotective pathways, such as HMOX1 upregulation that reduces ROS production or action. However, sustained upregulation of HMOX1 can lead to an over accumulation of Fe^{2+} , leading to ferroptosis mediated cell death. So even though these drugs may induce initial cytoprotection, they could subsequently lead to cell death and sustained lung injury. This is also reflected by the permeability data where lovastatin initially provides barrier protection but eventually shows an increase in permeability. Nevertheless, we agree that it is too early to conclude that the treatment with these drugs and the increased HMOX-1 expression mediate the suppression of acute radiation injury. Indeed, we have carried out additional studies that siRNA knock down of HMOX1 prevents some radiation induced lung responses, but not others, which are now included in new **Fig. 5m-o**.

10. The tissue hypoxia in the lung following irradiation is also a contributing factor to RILI. Can authors provide information on the oxygen levels observed in the Lung Alveolus Chip at various radiation doses?

We did not evaluate tissue hypoxia in the Lung Chip in this study; however, we carried out additional studies to assess the dose-dependent effects of radiation exposure on reactive oxygen species (ROS) levels in the alveolar epithelium and endothelium by using a fluorogenic probe (CellROX green) that detects oxidative stress in cells. This dye is cell-permeable and weakly fluorescent in a reduced state, whereas upon oxidation by ROS, it exhibits bright green fluorescence. We found that ROS levels increased as the radiation dose increased. The alveolar endothelium was particularly susceptible to radiation dosage with 21% of cells being CellROX+ at 8 Gy, indicating a significant increase in ROS levels in response to radiation. We now include these data in new **Supp. Fig. S1e** and describe them in the Results and Discussion.

11. Literature suggests that early acute radiation pneumonitis typically develops 2-12 weeks after radiotherapy. Consequently, the experimental timeframe of 6 hours to 7 days may not be adequate to fully evaluate the consequences of radiation exposure.

The central objective of this research was to recapitulate the effects of a single, high dose exposure to ionizing radiation during the acute phase of radiation injury in the human lung, which is not possible to measure directly in vivo. Moreover, the design of our model enables us to gain insight into the earliest responses to radiation injury which occur days before they are manifested in a way that can be detected clinically as pneumonitis. This is a key advantage of our system.

Our results show that the earliest response to radiation is the loss in barrier function, which results from high levels of ROS production in the endothelium and lower levels in the epithelium, which result in damage to cells within both tissues. The inflammatory response caused by this damage induces a cascade of inflammation involving increased cytokine release and vascular permeability within days or weeks, leading to symptoms as early as 1-3 weeks from radiation exposure. While clinically, these effects may take several weeks or months to manifest and be diagnosed, this organ chip model enables us to directly assess the entire natural history of this response and analyze its molecular basis during the earliest phase of this injury process. Finally, as we note in the Discussion, recent guidelines of the American Thoracic Society [15; References are listed at the end of this document] suggest that preclinical (animal) models of acute lung injury should mimic 3 out of 4 criteria: 1) histopathologic evidence of tissue injury, 2) alteration of the alveolar-capillary barrier, 3) an inflammatory response, and 4) physiological dysfunction. Thus, our in vitro preclinical *human* model captures these key hallmarks that are essential to qualify it as an experimental model for acute lung injury, while the Transwell model does not. Thus, we believe that the novelty and value of this model for the radiation injury field is clear. Longer-term effects of radiation can be studied in this model as well, but that would be the focus of a future study.

12. What concentrations of prednisolone and lovastatin were used in the study?

Prednisolone was used at a concentration of 200 ng/ μ L (C_{max} 113-1343 ng/ μ L) and lovastatin was used at its C_{max} concentration of 3.013 ng/ μ L.

13. What was the length of time that the chips were exposed to radiation within the range of 12-16 Gy during each exposure?

The chips were radiated at a dosage rate of 0.0133 Gy/s (15 min for 12 Gy and 20 min for 16 Gy exposures).

14. The legend of Figure 1D states that the quantification of 53BP1 foci was done per cell, but the figure shows the number of foci per nucleus. What methodology was utilized to determine the number of 53BP1 foci per nucleus/cell?

The measurement used was number of foci/nucleus. This error has been corrected.

REFERENCES

1. Arroyo-Hernández, M., et al., *Radiation-induced lung injury: current evidence*. BMC pulmonary medicine, 2021. **21**(1): p. 1-12.
2. Ghafoori, P., et al., *Radiation-induced lung injury: assessment, management, and prevention*. Oncology, 2008. **22**(1): p. 37.
3. Giuranno, L., et al., *Radiation-induced lung injury (RILI)*. Frontiers in oncology, 2019. **9**: p. 877-892.
4. Vozenin, M.-C., J.H. Hendry, and C. Limoli, *Biological benefits of ultra-high dose rate FLASH radiotherapy: sleeping beauty awoken*. Clinical oncology, 2019. **31**(7): p. 407-415.
5. Lee, Y., et al., *Therapeutic effects of ablative radiation on local tumor require CD8+ T cells: changing strategies for cancer treatment*. Blood, The Journal of the American Society of Hematology, 2009. **114**(3): p. 589-595.

6. Favaudon, V., et al., *Ultra-high dose-rate FLASH irradiation increases the differential response between normal and tumor tissue in mice*. *Science translational medicine*, 2014. **6**(245): p. 245ra93-245ra93.
7. Weichselbaum, R.R., et al., *Radiotherapy and immunotherapy: a beneficial liaison?* *Nature reviews Clinical oncology*, 2017. **14**(6): p. 365-379.
8. Adebahr, S., et al., *LungTech, an EORTC Phase II trial of stereotactic body radiotherapy for centrally located lung tumours: a clinical perspective*. *The British journal of radiology*, 2015. **88**(1051): p. 20150036.
9. Rosen, E.M., D.W. Vinter, and I.D. Goldberg, *Hypertrophy of cultured bovine aortic endothelium following irradiation*. *Radiation research*, 1989. **117**(3): p. 395-408.
10. Ungvari, Z., et al., *Ionizing radiation promotes the acquisition of a senescence-associated secretory phenotype and impairs angiogenic capacity in cerebrovascular endothelial cells: role of increased DNA damage and decreased DNA repair capacity in microvascular radiosensitivity*. *Journals of Gerontology Series A: Biomedical Sciences and Medical Sciences*, 2013. **68**(12): p. 1443-1457.
11. Yang, Y., et al., *ZBP1-MLKL necroptotic signaling potentiates radiation-induced antitumor immunity via intratumoral STING pathway activation*. *Science Advances*, 2021. **7**(41): p. eabf6290.
12. Takaoka, A., et al., *DAI (DLM-1/ZBP1) is a cytosolic DNA sensor and an activator of innate immune response*. *Nature*, 2007. **448**(7152): p. 501-505.
13. Liu, X., C. Shao, and J. Fu, *Promising Biomarkers of Radiation-Induced Lung Injury: A Review*. *Biomedicines*, 2021. **9**(9).
14. Choi, S.-H., et al., *A hypoxia-induced vascular endothelial-to-mesenchymal transition in development of radiation-induced pulmonary fibrosis*. *Clinical Cancer Research*, 2015. **21**(16): p. 3716-3726.
15. Kulkarni, H.S., et al., *Update on the features and measurements of experimental acute lung injury in animals: An official American Thoracic Society workshop report*. *American Journal of Respiratory Cell and Molecular Biology*, 2022. **66**(2): p. e1-e14.

REVIEWERS' COMMENTS

Reviewer #1 (Remarks to the Author):

The authors have carefully revised the manuscript and have responded to all criticism raised during the review. This is a clearly written manuscript of high quality, and the key statements are supported by experimental evidence. I think the manuscript can be accepted as is.

Reviewer #2 (Remarks to the Author):

The manuscript has been substantially improved, with toned down claims, new data, and clarified figures/text. It is now suitable for publication.

Reviewer #3 (Remarks to the Author):

I find that the author revision to my previous concerns is satisfactory. I do not have additional comments at this moment.